# Practical Shuffle Coding

**Julius Kunze**
University College London
juliuskunze@gmail.com

**Daniel Severo**
University of Toronto and Vector Institute
d.severo@mail.utoronto.ca

**Jan-Willem van de Meent**
University of Amsterdam
j.w.vandemeent@uva.nl

**James Townsend**
University of Amsterdam
j.h.n.townsend@uva.nl

## Abstract

We present a general method for lossless compression of unordered data structures, including multisets and graphs. It is a variant of shuffle coding that is many orders of magnitude faster than the original and enables 'one-shot' compression of single unordered objects. Our method achieves state-of-the-art compression rates on various large-scale network graphs at speeds of megabytes per second, efficiently handling even a multi-gigabyte plain graph with one billion edges. We release an implementation that can be easily adapted to different data types and statistical models.

## 1 Introduction

Big data is often contained in unordered objects, such as sets, multisets, graphs, or hypergraphs. Unlike ordered data types like text, audio, and video, the order of elements in these structures is irrelevant and represents redundant information. Recent work by Kunze et al. (2024) shows that eliminating this redundancy can lead to significant storage and transmission savings. It proposed shuffle coding, an entropy coding method based on bits-back (Townsend et al., 2019) with asymmetric numeral systems (ANS; Duda, 2009) that approaches optimal compression rates for sequences of unordered objects. However, their specific method, which we will refer to as complete joint shuffle coding, has two major limitations that make it impractical for large unordered data structures: It incurs a prohibitive initial bit cost in one-shot scenarios where only a single unordered object needs to be compressed, and it requires the exact computation of automorphism groups which is often slow or completely intractable, for example for large graphs.

To overcome these limitations, we propose two new variants, *autoregressive* and *incomplete* shuffle coding. Autoregressive shuffle coding builds on recent work by Severo et al. (2023a), which constructed an optimal one-shot codec for multisets from a codec of vectors by storing information in an ordering. This method depends on the simple structure of multisets' automorphism groups, and does not extend to other unordered objects such as unlabeled graphs. Autoregressive shuffle coding generalizes this method to arbitrary unordered objects. Incomplete shuffle coding approximates an object's symmetries, enabling compression despite intractable automorphism groups. These two variants can be combined into a method allowing one-shot compression of large unordered data structures at practical speeds.

Our experiments show that our high-performance implementation matches the optimal compression rates from Severo et al. (2023a) for multisets, but is orders of magnitude faster. Similarly, it is many orders of magnitude faster than joint shuffle coding for medium-sized graphs of various types with minimal rate increase. We also compress much larger unordered graphs, including social network graphs and random graphs with up to a billion edges, which are infeasible for complete shuffle

coding. We show that the rate discount from removing order information is large, as previously shown for joint shuffle coding on many smaller graph datasets, and that we achieve state-of-the-art graph compression rates at practical speeds when using models with minimal parameters. We release source code[1] which can be extended easily to new models and classes of unordered objects other than multisets and graphs.

The structure of the paper is as follows: Section 2 reviews the necessary background, including permutable classes, unordered objects and codecs. Sections 3 and 4 detail our incomplete and autoregressive shuffle coding variants. We discuss related work in Section 5, our experimental results in Section 6, and conclude in Section 7 with future research directions. Proofs are in Appendix B.

## 2 Background

Table 1: Examples for key concepts from Section 2 for the permutable class $\mathcal{F}$ of ASCII strings of length 5. $\{\!\{\ldots\}\!\}$ denotes a multiset, and we use cycle notation for permutations.

| Concept | Example |
|---------|---------|
| Permutation | $(0,2)(1,3) \in \mathcal{S}_5$ |
| Permutable class | $\mathcal{F} = \{\mathsf{aaaaa}, \mathsf{aaaab}, \ldots\}$ |
| Ordered object | $\mathsf{sense} \in \mathcal{F}$ $\qquad\qquad\qquad\qquad\qquad\qquad\qquad (0,2) \cdot \mathsf{sense} = \mathsf{nesse}$ |
| Quotient class | If $\mathsf{SENSE} \sim \mathsf{Sense} \sim \mathsf{sense} \sim \ldots$ ($\sim$ is equality up to case), $\mathsf{sense}_\sim = \{\mathsf{SENSE}, \mathsf{Sense}, \mathsf{sense}, \ldots\} \in \mathcal{F}_\sim$ $\quad (0,2) \cdot \mathsf{sense}_\sim = \mathsf{nesse}_\sim$ |
| Unordered object | $\overline{\mathsf{sense}} = \{\!\{\mathsf{s}, \mathsf{e}, \mathsf{n}, \mathsf{s}, \mathsf{e}\}\!\} = \{\mathsf{sense}, \mathsf{esnse}, \mathsf{ensse}, \mathsf{enses}, \mathsf{snese}, \ldots\} \in \overline{\mathcal{F}}$ |
| Automorphisms | $\mathrm{Aut}(\mathsf{sense}) = \{(0), (0,3), (1,4), (0,3)(1,4)\}$ |

This paper is concerned with compression of *unordered* objects. Kunze et al. (2024) define these in terms of equivalence classes which comprise ordered objects that are identical up to re-ordering. We introduce ordered and unordered objects in Section 2.1, as well as how equivalence classes of ordered objects can form ordered objects themselves, a central idea of this paper. We review codecs in Section 2.2 and the optimal compression rate for unordered objects in Section 2.3. We provide a summary of concepts through examples in Table 1.

### 2.1 Permutable classes

Concepts from group theory, including subgroups, actions, stabilizers, and orbits are used throughout. We provide a brief introduction in Appendix A.

For $n \in \mathbb{N}$, we let $[n] := \{0, 1, \ldots, n-1\}$, with $[0] = \emptyset$. For this paper, it is convenient to identify permutations purely by their cycle notation, such that we can apply a permutation of order $i$ to any objects of order $\geq i$. This can be achieved by defining the symmetric group $\mathcal{S}_n$ on the subset of bijections $f : \mathbb{N} \to \mathbb{N}$ with $f(i) = i$ for all $i \in \mathbb{N} \setminus [n]$, leading to

$$\mathcal{S}_i = \mathrm{Stab}_{\mathcal{S}_{i+1}}(i) \leq \mathcal{S}_{i+1}. \tag{1}$$

From here on, we use the shorthand $H \leq G$ to mean that $H$ is a subgroup of $G$. $\mathrm{Stab}_G(i)$ denotes the stabilizer with respect to $i \in [n]$ of a group $G \leq \mathcal{S}_n$ acting on the indices $[n]$.

We will consider objects which can be 're-ordered' by applying permutations. This is captured in the following definition from Kunze et al. (2024):

**Definition 2.1** (Permutable class). For $n \in \mathbb{N}$, a *permutable class* $\mathcal{F}$ of order $n$ is a pair of a set $F$ and a left group action of the permutation group $\mathcal{S}_n$ on $F$, which we denote with the $\cdot$ binary operator. We also use $\mathcal{F}$ to refer to the underlying set $F$. We refer to the elements of $F$ as *ordered objects*.

Strings are ordered objects, and so are labeled graphs because their vertices can be re-ordered. Equivalence classes of ordered objects can themselves form a permutable class. For example, case-insensitive ASCII strings naturally are ordered objects, and can be defined as the equivalence classes

---

[1]Source code, data and results are available at `https://github.com/juliuskunze/shuffle-coding`.

of such strings under an equivalence relation $\sim$ that indicates equality up to character case, as illustrated in Table 1. As shown in Appendix B.1, this happens whenever the equivalence relation is preserved under permutations, in the following sense:

**Definition 2.2** (Congruence, quotient class). We say an equivalence relation $\sim$ on a permutable class $\mathcal{F} = (F, \cdot)$ of order $n$ is a *congruence* if $s \cdot f \sim s \cdot g$ holds if $f \sim g$ for all $f, g \in \mathcal{F}$ and $s \in \mathcal{S}_n$. Then, the quotient set $F/\sim$ equipped with the operator $\cdot : \mathcal{S}_n \times (F/\sim) \to F/\sim$ with $s \cdot f_\sim := \{s \cdot f \mid f \in f_\sim\}$ forms a permutable class of order $n$, which we refer to as the *quotient class* of $\mathcal{F}$ by $\sim$, denoted as $\mathcal{F}/\sim$. For $f \in \mathcal{F}$ we use $f_\sim$ to denote the equivalence class under $\sim$ containing $f$.

We define unordered objects as equivalence classes comprising ordered objects that are identical up to re-ordering, an important special case of Definition 2.2:

**Definition 2.3** (Isomorphism, unordered objects). For two objects $f$ and $g$ in a permutable class $\mathcal{F}$, we say that $f$ is *isomorphic* to $g$, and write $f \simeq g$, if there exists $s \in \mathcal{S}_n$ such that $g = s \cdot f$. The isomorphism relation $\simeq$ is a congruence, inducing a quotient class of *unordered objects* that we will denote as $\overline{\mathcal{F}} := \mathcal{F}/\simeq$, and the unordered object containing some $f \in \mathcal{F}$ as $\bar{f} := f_\simeq$.

Unordered strings then correspond to multisets (for example $\overline{\mathsf{see}} = \{\!\{\mathsf{e}, \mathsf{e}, \mathsf{s}\}\!\} = \{\mathsf{see}, \mathsf{ese}, \mathsf{ees}\}$) and unordered graphs to unlabeled graphs. The set of permutations that, when applied to an ordered object, do not change it, forms a group indicating its 'symmetries':

**Definition 2.4** (Automorphism group). For an element $f$ of a permutable class $\mathcal{F}$, we let $\mathrm{Aut}(f)$ denote the *automorphism group* of $f$, defined by $\mathrm{Aut}(f) := \{s \in \mathcal{S}_n \mid s \cdot f = f\}$.

For the example of unordered objects, every element's automorphism group comprises all permutations, $\mathrm{Aut}(\bar{f}) = \mathcal{S}_n$.

## 2.2 Codecs

Shuffle coding requires stack-like (LIFO) codecs, such as those based on the range variant of asymmetric numeral systems (rANS; Duda, 2009), to save bits corresponding to the redundant order using bits-back (Townsend et al., 2019). To define these, we fix a set $M$ of prefix-free binary messages, and a length function $l \colon M \to [0, \infty)$, which measures the number of physical bits required to represent values in $M$. We rely on the following definition from Kunze et al. (2024):

**Definition 2.5** (Codec). A *stack-like codec* (or simply *codec*) for a set $X$ is an invertible function encode $: M \times X \to M$. We call a codec *optimal* for a probability distribution over $X$ with mass function $P$ if for any $m \in M$ and $x \in X$, the message length $l$ satisfies[2] $l(\mathrm{encode}(m, x)) \approx l(m) + \log \frac{1}{P(x)}$. We refer to $\log \frac{1}{P(x)}$ as the *optimal rate* and to the inverse of encode as decode. Since decode has to be implemented in practice, we treat it as an explicit part of a codec below.

The encode function requires a pre-existing message as its first input. Therefore, at the beginning of encoding we set $m$ equal to some fixed, short initial message $m_0$, with length less than 64 bits. As in other entropy coding methods, which invariably have some small constant overhead, this 'initial bit cost' is amortized as we compress more data.

## 2.3 Optimal rate for unordered objects

Our codec will be based on a probability distribution with mass function $P$ over ordered objects from a permutable class $\mathcal{F}$. Then, for any equivalence relation $\sim$ on $\mathcal{F}$, a joint distribution $P(f_\sim, g)$ with equivalence classes $f_\sim \in \mathcal{F}/\sim$ is induced by sampling an ordered object $g \in \mathcal{F}$ from $P$, and returning its corresponding equivalence class, resulting in

$$P(f_\sim) = \sum_{g \in f_\sim} P(g). \tag{2}$$

---

[2]This condition, with a suitable definition of $\approx$, is equivalent to rate-optimality in the usual Shannon sense, see Townsend (2020).

We will implicitly assume this induced distribution over quotient classes, even in nested cases. Kunze et al. (2024) specify the optimal rate of any codec for unordered objects $\bar{f}$,

$$\log \frac{1}{P(\bar{f})} = \underbrace{\log \frac{1}{P(f)}}_{\text{Ordered rate}} - \underbrace{\log \frac{n!}{|\text{Aut}(f)|}}_{\text{Discount}},$$ (3)

assuming, without loss of modeling power, that $P(f)$ is exchangeable, meaning invariant under permutations of $f$.

## 3 Incomplete shuffle coding

Table 2: Example for color refinement as defined in Appendix C on plain graphs with $0$ and $1$ convolutions, and the resulting incompletely ordered graphs according to Definition 3.1. Colors only visualize correspondence to the resulting hashes, they are not materialized in the graph. The actual hashes, characters in the example, depend on the specific hash function used. $C_0$ bases each vertexes' hash on its degree only, while $C_1$ also takes the multiset of neighboring hashes from $C_0$ into account. The automorphism group of the incompletely ordered graph is the same as that for the string of vertex hashes, and color partitions correspond to their orbits.

| Coloring | Incompletely ordered graph |
|---|---|
| $C_0 \left( \begin{smallmatrix} 2 \\ 1 \ 3\text{-}4 \\ 0 \end{smallmatrix} \right) = \texttt{aaacd}$ | $\left( \begin{smallmatrix} 2 \\ 1 \ 3\text{-}4 \\ 0 \end{smallmatrix} \right)_{\sim_{C_0}} = \left\{ \begin{smallmatrix} 2 \\ 1 \ 3\text{-}4 \\ 0 \end{smallmatrix} , \begin{smallmatrix} 2 \\ 1\text{+}3\text{-}4 \\ 0 \end{smallmatrix} , \begin{smallmatrix} 2 \\ 1\text{+}3\text{-}4 \\ 0 \end{smallmatrix} \right\}$ |
| $C_1 \left( \begin{smallmatrix} 2 \\ 1 \ 3\text{-}4 \\ 0 \end{smallmatrix} \right) = \texttt{babcd}$ | $\left( \begin{smallmatrix} 2 \\ 1 \ 3\text{-}4 \\ 0 \end{smallmatrix} \right)_{\sim_{C_1}} = \left\{ \begin{smallmatrix} 2 \\ 1 \ 3\text{-}4 \\ 0 \end{smallmatrix} \right\}$ |

Complete joint shuffle coding, as presented in Kunze et al. (2024), relies on a function to retrieve (a list of generators of) the automorphism group $\text{Aut}(f)$, as well as the canonization for any given ordered object $f$. No polynomial-time algorithm is known to compute this function for graphs.[3] Accordingly, their results show that the method is impractically slow even for moderately sized graphs.

In this section, we will introduce a method that gives shuffle coding a reliably fast runtime for graphs. To achieve this, we will in return accept slightly suboptimal compression rates. Instead of recovering the exact order and therefore realizing the complete bit discount, we can ignore some hard-to-compute information about the order to improve runtime. Specifically, we can treat isomorphic objects that are hard to distinguish as elements of the same 'incompletely ordered' object, formalized as follows:

**Definition 3.1** (Incompletely ordered objects). We refer to the elements of a quotient class $\mathcal{F}/\sim$ as *incompletely ordered objects* if $f_\sim \subseteq \bar{f}$ for all $f \in \mathcal{F}$.

We show in Appendix B.3 that their unordered objects $\overline{\mathcal{F}/\sim}$ correspond to the original unordered objects $\overline{\mathcal{F}}$ through the bijection $\bar{f} = \bigcup \overline{f_\sim}$, leading to

$$\log \frac{1}{P(f)} - \log \frac{1}{P(f_\sim)} = \log \frac{|\text{Aut}(f_\sim)|}{|\text{Aut}(f)|}$$ (4)

Equation (4) reveals the rate increase compared to the optimal rate if we suboptimally code $f_\sim$ by representing it by an arbitrary $f \in f_\sim$ and using a codec optimal for $P(f)$.

We can apply joint shuffle coding to incompletely ordered objects, an approach we refer to as *incomplete joint shuffle coding*. We are then not recovering the 'hard-to-compute' bits to distinguish

---

[3]Libraries commonly used in practice such as `nauty` and `Traces` (McKay and Piperno, 2014) have exponential worst-case runtime. An algorithm with quasi-polynomial runtime complexity is known (Babai, 2016, 2019; Helfgott et al., 2017), but is so complicated that it has never been implemented.

elements within $f_\sim$, leading to a rate increase given by Equation (4) over the optimal rate. This requires two functions determining the automorphism group $\mathrm{Aut}(f_\sim)$ and a canonization $\hat{f}_\sim$ for any given such $f_\sim$. In the next section, we introduce a choice for $\sim$ that makes these functions and therefore incomplete joint shuffle coding practical for graphs.

Any function $C$ on $\mathcal{F}$ with the relation defined by $C(f) = C(g)$ being a congruence has an associated class of incompletely ordered objects $\mathcal{F}/\sim_C$ through $f \sim_C g$ exactly if $f \simeq g$ and $C(f) = C(g)$. The *color refinement algorithm*, also known as the 1-dimensional version of the algorithm of Leman and Weisfeiler (1968), is a practical choice for such a function for graphs. It returns a string of $n$ vertex 'colors' that were iteratively refined by hashing local features through a graph convolution, starting from the vertex degrees, as visualized in Table 2, and formalized in Appendix C. As discussed there, any $k > 0$ convolutions suffice to find the exact automorphism group $\mathrm{Aut}(g_{\sim_{C_k}}) = \mathrm{Aut}(g)$ for almost all simple graphs for large enough $n$ (Babai et al., 1980), and thus in practice the compression rate of incomplete shuffle coding with color refinement is usually optimal or near-optimal.

When used for incomplete joint shuffle coding, color refinement yields a string of vertex 'colors' that assigns a graph to a specific incomplete ordered graph $g_{\sim_{C_k}}$. Since these vertex colors form a string, we can apply complete joint shuffle coding which is fast for multisets, to approximately canonize graphs and code string cosets. As shown in Appendix B.3, for a fixed number of convolutions $k$, the resulting, much improved overall runtime complexity is

$$O(m + n \log n), \tag{5}$$

where $m$ denotes the number of edges in the graph.

## 4  Autoregressive shuffle coding

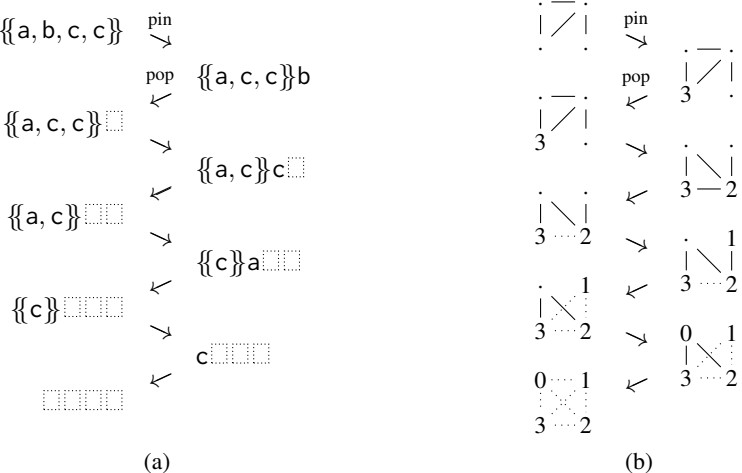

(a)                                                                 (b)

Figure 1: Iterations for autoregressive shuffle coding during encoding of (a) a multiset and (b) an unlabeled undirected graph. Dotted placeholders indicate deleted information. Decoding an orbit allows to 'pin' an element in the last position. The pinned element is subsequently 'popped' from the object and encoded, and the process is repeated recursively on the remaining unordered prefix.

Joint shuffle coding, both complete and incomplete, incurs a prohibitive initial bit cost in one-shot scenarios where only a single unordered object needs to be compressed. This is because it constructs an encoder by *decoding* all order information from the message (the bits-back step) before the ordered object is encoded with a given 'ordered' codec optimal for $P(f)$. Therefore, other information has to be encoded into the message $m$ before an unordered object. At the very beginning of encoding, these 'initial bits' can be generated at random, but they are unavoidably encoded into the message. While for sequences of such objects, this constant initialization overhead is amortized and the rate tends to the optimal rate with more objects being compressed, it means that joint shuffle coding realizes no discount when coding a single unordered object, rendering it useless in the one-shot case.

Specifically, joint shuffle coding realizes the discount from Equation (3) in the net rate completely (or incompletely if using the variant from Section 3), but none of it in the (one-shot) rate. We now

aim to realize most of the discount in the rate by progressively decoding the permutation while autoregressively encoding the object. Severo et al. (2023a) implement this idea for the special case of multisets, as visualized in Figure 1(a). However, it exploits the simple structure of a string's automorphism group and does not extend to other unordered objects such as graphs. In this section, we will generalize this codec to arbitrary unordered objects, including graphs, as shown in Figure 1(b).

Table 3: Examples of key concepts from Section 4 for the permutable class $\mathcal{F}$ of ASCII strings of length 5 and the prefixing chain given by Example 4.1. $\{\!\{\ldots\}\!\}$ denotes a multiset, and ⬚ indicates an arbitrary ASCII character. We visualize the same concepts for graphs in Table 4 in Appendix D.

| Concept | Example |
|---------|---------|
| Unordered derivative | $\overline{\text{sense}'} = \{\!\{\text{s}, \text{e}, \text{n}, \text{s}\}\!\}\text{e} = \{\text{ensse}, \text{esnse}, \ldots, \text{ssnee}\} \in \overline{\mathcal{F}'}$ |
| Unordered $i$-th derivative | $\overline{\text{sense}^{(3)}} = \{\!\{\text{s}, \text{e}\}\!\}\text{nse} = \{\text{sense}, \text{esnse}\} \in \overline{\mathcal{F}^{(3)}}$ |
| Prefix | $\text{sense}_{[3]} = \text{sen}\square\square = \{\text{senaa}, \text{senab}, \ldots, \text{senba}, \text{senbb}, \ldots\} \in \mathcal{F}_{[3]}$ |
| Slice | $\text{sense}_2 = \{\!\{\text{s}, \text{e}\}\!\}\text{n}\square\square$ , represented by $\text{n}$ in context of $\{\!\{\text{s}, \text{e}\}\!\}\square\square\square$ |
| Pop | $\text{pop}(\text{sen}\square\square) = (\text{se}\square\square\square, \{\!\{\text{s}, \text{e}\}\!\}\text{n}\square\square)$ |
| Push | $\text{push}((\text{se}\square\square\square, \{\!\{\text{s}, \text{e}\}\!\}\text{n}\square\square)) = \text{sen}\square\square$ |
| Orbit of index $i$ | $\text{Orb}_{\text{Aut}(\text{sense})}(1) = \{1, 4\}$ |
| Orbits | $\text{Orbs}_{\text{Aut}(\text{sense})} = \{\{0, 3\}, \{1, 4\}, \{2\}\}$ |
| Orbit function | $\text{orbits}(\text{sense}) = (2, 0, 1, 2, 0)$ implies $\text{orbits}(\text{eenss}) = (0, 0, 1, 2, 2)$ |

## 4.1 Prefixes of ordered objects

A general notion of 'pinning' objects such as graphs, as visualized in Figure 1, can be formalized by disallowing any permutations involving the last position, exploiting Equation (1):

**Definition 4.1** (Derivative). For a permutable class $\mathcal{F} = (F, \cdot)$ of order $n > 0$, the pair of $F$ and $\cdot' : \mathcal{S}_{n-1} \times F \to F$ with $s \cdot' f := s \cdot f$ forms a permutable class of order $n - 1$, which we refer to as the *derivative* of $\mathcal{F}$, denoted as $\mathcal{F}'$. We say that the unordered object of this class containing $f$ is the *unordered derivative* of $f$, denoted as $\overline{f'}$. By applying the derivative $i \in [n]$ times, we obtain a permutable class of order $n - i$, which we refer to as the $i$-th derivative of $\mathcal{F}$, denoted as $\mathcal{F}^{(i)}$. We define the unordered $i$-th derivative of $f$ in the same way, denoted as $\overline{f^{(i)}}$.

The derivative $\mathcal{F}'$ has the same elements as $\mathcal{F}$, but its unordered objects $\overline{f'}$ are different: $\overline{\text{seen}'} = \{\!\{\text{s}, \text{e}, \text{e}\}\!\}\text{n} = \{\text{seen}, \text{esen}, \text{eesn}\} \neq \overline{\text{seen}}$. While the derivative $\mathcal{F}'$ pins the last position, the $i$-th derivative $\mathcal{F}^{(i)}$ pins the last $i$ positions.

Length $i$ string prefixes are ordered objects of order $i$, which we formalize as follows:

**Example 4.1** (String prefixes). For $i \in [n + 1]$ and a set of elements $X$, let $\sim_{[i]}$ be the equivalence relation on strings $X^n$ with $f \sim_{[i]} g$ exactly if the first $i$ elements of $f$ and $g$ are equal for $f, g \in X^n$. We denote the equivalence class of $f \in X^n$ under $\sim_{[i]}$ as $f_{[i]}$ and refer to it as (string) prefix of $f$ of length $i$. While $\sim_{[i]}$ is not a congruence on $X^n$, it is on the $(n-i)$-th derivative $(X^n)^{(n-i)}$, and $(X^n)^{(n-i)}/\sim_{[i]}$ therefore forms a quotient class.

While a string naturally has 'prefixes' and 'elements', it is not obvious what the equivalent notions should be for graphs, and more generally, ordered objects. The conditions our method requires are captured in the following generalized definition of prefixes, for which Example 4.1 is a special case. Importantly, the notion of a 'slice' will be used in place of 'string element', as visualized in Table 3:

**Definition 4.2** (Prefixes of ordered objects). For a permutable class $\mathcal{F}$ of order $n$ and all $i \in [n + 1]$, let $\sim_{[i]}$ be a congruence on $\mathcal{F}^{(n-i)}$. We then refer to the tuple of quotient classes $\mathcal{F}_{[i]} := \mathcal{F}^{(n-i)}/\sim_{[i]}$ as a *prefixing chain* on $\mathcal{F}$ if the equivalence class of $f \in \mathcal{F}$ under $\sim_{[i]}$, referred to as *prefix* of $f$ of length $i$ and denoted as $f_{[i]}$, and $f_i := \bigcup \overline{f'_{[i+1]}}$, referred to as the *slice* of $f$ at index $i$, fulfill

$$f_{[n]} = \{f\}, \tag{6}$$

meaning that an object of order $n$ is uniquely determined by its prefix of length $n$, and

$$f_{[i+1]} = f_{[i]} \cap f_i, \tag{7}$$

meaning that a prefix of length $i + 1$ is uniquely determined by the prefix of length $i$ and the slice at index $i$. The latter condition ensures the existence of an invertible function $\mathrm{pop} : \mathcal{F}_{[i+1]} \to \{(f_{[i]}, f_i) \mid f \in \mathcal{F}\}$ with $\mathrm{pop}(f_{[i+1]}) = (f_{[i]}, f_i)$. We will denote its inverse as $\mathrm{push}$.

For strings, the slice $f_i$ uniquely determines the element at index $i$. In practice, we can represent a string slice $f_i$ simply as the string element at index $i$ since we only ever use it in context of a prefix $f_{[i]}$. This definition of prefixes is general enough to apply to graphs, , as visualized in Table 4:

**Example 4.2** (Graph prefixes). For $i \in \{0, \ldots, n\}$, let $\sim_{[i]}$ be the relation on simple graphs $\mathcal{G}_n$ with $f \sim_{[i]} g$ exactly if $f$ and $g$ are equal up to the edges between the last $n - i$ vertices for $f, g \in \mathcal{G}_n$. The quotient classes $\mathcal{G}_n^{(n-i)}/\sim_{[i]}$ then form a prefixing chain. Graph prefixes $f_{[i]}$ can then be represented as graphs without any edges between the last $n - i$ vertices. Because a graph slice $f_i$ for $i \in [n]$ is only ever used in context of a prefix $f_{[i]}$, we represent it as the subset of edges from $\{(i, i+1), (i, i+2), \ldots, (i, n-1)\}$ in practice.

Given a prefixing chain and an exchangeable probability distribution $P$ over $\mathcal{F}$, the optimal rate for an unordered prefix $\overline{f_{[i]}}$ is

$$\log \frac{1}{P(\overline{f_{[i]}})} = \log \frac{1}{P(\overline{g_{[i-1]}})} + \underbrace{\log \frac{1}{P(g_{i-1} \mid g_{[i-1]})}}_{\text{Slice rate}} - \underbrace{\log \frac{i}{|\mathrm{Orb}_{\mathrm{Aut}(f_{[i]})}(j)|}}_{\text{Orbit discount}}, \tag{8}$$

where $g_{[i]} = (j, i - 1) \cdot f_{[i]}$ for any $j \in [i]$. For proof see Appendix B.2.

## 4.2 Achieving the target rate

The recursive structure in Equation (8) hints at how to construct a recursive *codec* for unordered prefixes optimal for $P(\overline{f_{[i]}})$. To obtain $g_{[i]}$, we require a function $\mathrm{swap}(f_{[i]}, j, k) := (j, k) \cdot f_{[i]}$ that swaps two positions of a prefix $f_{[i]}$. To realize the slice rate term from Equation (8), we require an (autoregressive) codec for slices $f_i$ parameterized by $g_{[i]}$ that is optimal for $P(f_i \mid g_{[i]})$, denoted as $\mathsf{Slice}$. For simplicity, we assume that there is only a single unique 'empty' prefix of length 0, denoted as $\mathsf{f0}$ below, as is the case for Examples 4.1 and 4.2. We will realize the orbit discount by recovering the bits for the orbit of $j$ in $\mathrm{Aut}(f_{[i]})$. For that, we need a way to identify the orbits of any $f \in \mathcal{F}$ in a way that is invariant under permutations, formalized through the following definition:

**Definition 4.3** (Orbit function). For a permutable class $\mathcal{F}$ of order $n$, let $\leq$ be a total order on the orbits $O = \mathrm{Orbs}_{\mathrm{Aut}(f)} := \{\mathrm{Orb}_{\mathrm{Aut}(f)}(i) \mid i \in [n]\}$ of all indices $i \in [n]$. This induces a unique function $I : O \to [|O|]$ with $I(o) \leq I(o')$ exactly if $o \leq o'$ for all $o, o' \in O$. We then refer to the function $\mathrm{orbits}(f) := \left( I(\mathrm{Orbs}_{\mathrm{Aut}(f)}(j)) \right)_{j \in [n]}$ for all $f \in \mathcal{F}$ as an *orbit function* of $\mathcal{F}$ if $\mathrm{orbits}(s \cdot f)_{s \cdot j} = \mathrm{orbits}(f)_j$ for all $s \in \mathcal{S}_i$, $f \in \mathcal{F}$ and $j \in [n]$.

An orbit function for string prefixes ranks its elements according to some order over the alphabet, as visualized in Table 3. For graph prefixes $f_{[i]}$, we can implement an orbit function by distinctly coloring the last $n - i$ vertices of the graph $f$, deleting the edges between them, and passing it to the $\mathsf{nauty}$ and $\mathsf{Traces}$ library (McKay and Piperno, 2014), which provides an orbit function for graphs.

We require an orbit function $\mathrm{orbits}_i$ on prefixes $\mathcal{F}_{[i]}$ of all lengths $i \in [n]$. This defines a corresponding probability distribution over orbit indices $[|\mathrm{Orbs}_{\mathrm{Aut}(f_{[i]})}|]$ with mass function $P_o$ s.t. for all $i \in [n]$ and $j \in [i]$, $P_o(\mathrm{orbits}_i(f_{[i]})_j)) := |\mathrm{Orb}_{\mathrm{Aut}(f_{[i]})}(j)|/i$. We construct a codec $\mathsf{Orbit}$ optimal for $P_o$ by using a categorical codec with masses proportional to the counts of each orbit index $\mathrm{orbits}_i(f_{[i]})$.

We list the autoregressive codec $\mathsf{UnorderedPrefix}$, parameterized by the prefix length $i$:

```python
def encode(m, f):
    if i=0: return m
    os = orbits(f)
    m, o = Orbit(os).decode(m)

    j = find(o, os)
    g = swap(f, j, i-1)
    g1, s = pop(g)
    m = Slice(g1).encode(m, s)
    m = UnorderedPrefix(i-1).encode(m, g1)

    return m

def decode(m):
    if i=0: return m, f0
    m, g1 = UnorderedPrefix(i-1).decode(m)
    m, s = Slice(g1).decode(m)
    g = push(g1, s)
    os = orbits(g)
    o = os[i-1]
    m = Orbit(os).encode(m, o)
    return m, g
```

Effect on message length:

$$-\log \frac{i}{|\mathrm{Orb}_{\mathrm{Aut}(f_{[i]})}(j)|}$$

$$+\log \frac{1}{P(g_i|g_{[i-1]})}$$
$$+\log \frac{1}{P(\overline{g}_{[i-1]})}$$

Similarly to joint shuffle coding, we represent an unordered prefix $\overline{f_{[i]}}$ by an arbitrary (ordered) prefix $f_{[i]} \in \overline{f_{[i]}}$, denoted as f in the listing. In the encoder, we decode an orbit index o from the message m according to $P_o$, and let $j$ be an arbitrary index within the corresponding orbit. We express this with a function find(o, os) which returns the first index of a given element in a given tuple. Equation (6) ensures that we can represent ordered objects $f \in \mathcal{F}$ as their prefixes $f_{[n]} = \{f\}$ of length $n$. This allows us to recursively code unordered objects $\bar{f}$ with UnorderedPrefix(n) optimal for $P(\bar{f})$, a method which we will refer to as *autoregressive shuffle coding*.

### 4.3 Achieving practical speeds

A naive implementation of autoregressive shuffle coding as shown in the listing above is impractically slow on larger objects: for every prefix length $i \in [n]$, the prefix orbits and the corresponding orbit codec (as well as the slice codec), need to be recomputed. In the case of graphs, finding the exact orbits for even one such graph prefix is slow, just like finding the automorphism group of a graph for complete joint shuffle coding. This subsection presents three techniques critical to achieving practical speeds with autoregressive shuffle coding.

**Adaptive entropy coding.** Instead of independently computing a new Slice and Orbit codec for each iteration using orbits, as shown in the code listing for simplicity, we allow reuse between iterations by updating their state during push, pop and swap. This allows to implement 'adaptive entropy coding', used by Severo et al. (2023a) in the context of Orbit for multisets $\overline{X^n}$, which achieves fast updates of a mutable categorical codec based on an order statistic tree that is weighted in the sense that it allows each element to be present some $k \in \mathbb{N}$ times. We will reuse the same approach to efficiently implement autoregressive Slice codecs, where for multisets we use a custom weighted AVL tree (Adelson-Velskii, 1962).

**Incomplete autoregressive shuffle coding.** Instead of computing the orbit function on prefixes $f_{[i]}$, we can apply it to some incompletely ordered version $(f_{[i]})_\sim$ instead, as defined in Definition 3.1. Similarly to Equation (4), this reduces the realized discount from Equation (8), increasing the rate by

$$\log \frac{|\mathrm{Orb}_{\mathrm{Aut}((f_{[i]})_\sim)}(j)|}{|\mathrm{Orb}_{\mathrm{Aut}(f_{[i]})}(j)|}, \tag{9}$$

bits in each recursive step. We refer to this variant as *incomplete autoregressive shuffle coding*. We can apply color refinement to graph prefixes by distinctly coloring the last $n - i$ vertices of the graph $f$, deleting the edges between them, and then applying it to graphs as usual (discarding the last $n - i$ vertex hashes). Applying incomplete autoregressive shuffle coding with color refinement by using the graph prefixing chain from Example 4.2 greatly improves runtime complexity compared to the

complete version. However, it still results in a runtime of at least $\Omega(ne)$ that is impractical for larger graphs, since color refinement has to be run for every prefix $f_{[i]} \in [n]$ times. Thus, it would be helpful to reduce the number of required color refinement runs to reduce runtime further, motivating the next technique.

**Chunking.** Instead of encoding order information after every slice, we can decode slices in a small fixed number $c$ of *chunks*, with a sequence $C$ of sizes adding up to $n$. For graphs, we can apply color refinement to the graph prefix after decoding the next chunk, and iteratively code the new order information according to the resulting vector of vertex colors. We then repeat this process until the complete graph is decoded. In Appendix E we formalize this generalized approach as (incomplete) autoregressive shuffle coding with a 'chunked' prefixing chain, where prefixes code more slices than usual to complete the respective chunk. For graphs, we then require only $c$ color refinement runs (instead of $n$ for the 'full' prefixing chain), resulting in a practical runtime of

$$O(c \cdot m + n \log n) \tag{10}$$

Coarser chunks generally lead to better runtimes but also higher initial bits overhead. The extreme case of using only a single chunk of size $n$ leads to a method sharing many properties with joint shuffle coding. Chunking allows using a Fenwick tree (Ryabko, 1989) for adaptive entropy coding since all possible values for vertex hashes in a chunk are known in advance, leading to better memory layout and faster runtimes. For a detailed discussion, see Appendix E.

## 5 Related work

**Complete joint shuffle coding.** Unlike joint shuffle coding introduced in Kunze et al. (2024), autoregressive shuffle coding allows one-shot compression of unordered objects. This is achieved by interleaving encoding and decoding steps while coding a single object, an approach first proposed in the context of 'Bit-Swap' (Kingma et al., 2019). Our method requires an autoregressive model for slices instead of a joint model. Unlike complete shuffle coding, incomplete shuffle coding does not require the automorphism group of an object and therefore does not depend on libraries such as `nauty` and `Traces` (McKay and Piperno, 2014).

**Graphs.** Choi and Szpankowski (2012) present a compression method called 'structural ZIP' (SZIP), which asymptotically achieves the rate

$$\log \frac{1}{P_{\text{ER}}(g)} - n \log n + O(n), \tag{11}$$

where $P_{\text{ER}}$ is the Erdős-Rényi $G(n, p)$ model. Compared to our method, SZIP is less flexible in the sense that it only applies to simple graphs (without vertex or edge attributes), and it is not an entropy coding method, thus the model $P_{\text{ER}}$ cannot be changed easily. The 'Partition and Code' (PnC; Bouritsas et al., 2021) method uses neural networks to compress unordered graphs. Unlike PnC, we achieve state-of-the-art compression rates when using simple models with minimal parameters, which amortize better when compressed along with a single graph.

**Multisets.** Our method generalizes Severo et al. (2023a) from multisets to arbitrary unordered objects, including graphs.

## 6 Experiments

To demonstrate our methods experimentally, we applied them to multisets and graphs. We report multiset compression results in Appendix G where we compare our implementation of full autoregressive shuffle coding to Severo et al. (2023a) on multisets of varying lengths, showing that while (one-shot) rates are matched, our implementation is well over two orders of magnitude faster and scaling to large multisets. Further multiset experiments show that joint shuffle coding is even faster while achieving the same net rates.

For graphs we applied incomplete joint and autoregressive shuffle coding with color refinement according to Definition C.1 to various graph datasets. We use the simple Erdős-Rényi (ER) $G(n, p)$ model for $P$, which is straightforward to convert into an autoregressive model $P(f_i \mid f_{[i]})$. Kunze et al. (2024) observe that the Pólya urn (PU) preferential attachment model proposed by Severo

et al. (2023b) drastically improves the compression rate on SZIP graphs from Choi and Szpankowski (2012). Motivated by this, we also use a variational approximation to obtain a tractable autoregressive model that we refer to as autoregressive Pólya urn (AP), described in Appendix F. We favored AP for larger graphs over PU because AP scaled better in terms of runtime, with similar rates.

**Joint.** We report results for incomplete joint shuffle coding on graphs in Appendix H. We first apply it to SZIP graphs with varying $k$ and report how this affects compression rate in Figure 4, to find a practical number of convolutions $k$. The results show that $k = 3$ is sufficient to get close to the optimal rate, which we therefore use for all other graph experiments in this paper. In Tables 8 and 9, we compare complete and incomplete joint shuffle coding on the TU (Morris et al., 2020) and SZIP datasets.[4] We observe that incomplete shuffle coding leads to dramatic speedups of up to a factor of one million on some of these graphs, with a minimal increase in compression rate across all datasets. We also evaluate incomplete joint shuffle coding using the AP codec on the large graphs used by Severo et al. (2023b) in Table 10, which we refer to as the 'REC' graph dataset, where complete joint shuffle coding is too slow to finish on any graphs. In the same table, we report compression speeds when run on a single thread and 8 threads, both on the SZIP and REC graphs, confirming an advantage with multiple threads.

**Autoregressive.** For incomplete autoregressive graph shuffle coding, we apply chunking to the prefixing chain from Example 4.2. We first evaluate the effect of chunk size in Appendix I on SZIP graphs, and find that $c = 16$ uniformly sized chunks lead to a good balance between runtime and rate, which we will use in all following experiments.

We apply incomplete autoregressive shuffle coding to the SZIP graphs based on ER and AP, and compare it to the SZIP method in Table 11. The results show that the advantage of a preferential attachment model over Erdős-Rényi carries over from the joint PU codec to the autoregressive approximation AP, demonstrating the value of shuffle coding being an entropy coder where the model can be changed easily. This results in significantly better rates compared to SZIP, at practical compression speeds that appear to be largely independent of graph size, whereas speeds for the SZIP codec seem to drop with graph size (SZIP graph sizes are shown in Table 10).

We evaluate autoregressive shuffle coding on REC graphs in Table 12, and compare it to the ordered rate without shuffle coding, demonstrating its practicality and rate advantage on large graphs. These results also confirm that the discount unrealized due to initial bits, visible as the difference between net rate and rate, is relatively small, highlighting the effectiveness of autoregressive shuffle coding for one-shot compression.

Finally, we evaluate autoregressive shuffle coding with the AP model on very large plain random graphs drawn from $G(n, m)$ Erdős-Rényi models and report rates and speeds in Table 13. The largest graph has one billion edges and an uncompressed size of 3.4 gigabytes, with our compression method saving 0.8 gigabytes. The speed results confirm the near-linear runtime predicted by Equation (10) across many orders of magnitude.

## 7  Conclusion

We proposed a general entropy coding method for unordered objects, achieving state-of-the-art compression rates at practical speeds for multisets and graphs up to gigabyte-scale. It is a combination of two new shuffle coding variants, as summarized in Table 5.

We believe that there is significant room for optimization. On REC graphs, we observed that roughly half of the runtime is spent on the ordered autoregressive model which has not yet been optimized. Chunking invites parallelization, for example with vectorized ANS (Giesen, 2014). Non-uniform chunk sizes $C$ are likely also beneficial for graphs, as already demonstrated for multisets in Appendix G. Unlike competing methods like SZIP and PnC, shuffle coding is an entropy method that can be easily adapted to specific domain models, and benefits from advances in generative graph modeling, such as recent work on neural (autoregressive) graph models (Kong et al., 2023; Zhu et al., 2022). We leave these promising research directions for future work.

---

[4]The TU dataset features vertex and edge attributes. Here, maximum-likelihood parameters for categorical attribute distributions are inferred for and coded along with each dataset, as described in Kunze et al. (2024).

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

# A Group actions, orbits and stabilizers

This appendix gives the definitions of group actions, orbits and stabilizers as well as a statement and proof of the orbit-stabilizer theorem, which we make use of in Section 4. We use the shorthand $H \leq G$ to mean that $H$ is a subgroup of $G$, and for $g \in G$, we use the usual notation, $gH := \{gh \mid h \in H\}$ and $Hg := \{hg \mid h \in H\}$ for left and right cosets, respectively. For any two subgroups $G, H \leq S$, we denote the group of their intersection as $G \cap H$.

**Definition A.1** (Group action). For a set $X$ and a group $G$, a *group action*, or simply *action*, is a binary operator

$$\cdot_G : G \times X \to X \tag{12}$$

which respects the structure of $G$ in the following sense:

1. The identity element $e \in G$ is neutral, that is $e \cdot_G x = x$.

2. The operator $\cdot_G$ respects composition. That is, for $g, h \in G$,

$$g \cdot_G (h \cdot_G x) = (gh) \cdot_G x. \tag{13}$$

We will often drop the subscript $G$ and use infix $\cdot$ alone where the action is clear from the context.

**Definition A.2** (Orbit). An action of a group $G$ on a set $X$ induces an equivalence relation $\sim_G$ on $X$, defined by

$$x \sim_G y \quad \text{if and only if there exists} \quad g \in G \quad \text{such that} \quad y = g \cdot x. \tag{14}$$

We refer to the equivalence classes induced by $\sim_G$ as *orbits*, and use $\mathrm{Orb}_G(x)$ to denote the orbit containing an element $x \in X$. We use $X/G$ to denote the set of orbits, so for each $x \in X$, $\mathrm{Orb}_G(x) \in X/G$.

**Definition A.3** (Stabilizer subgroup). For an action of a group $G$ on a set $X$, for each $x \in X$, the *stabilizer*

$$\mathrm{Stab}_G(x) := \{g \in G \mid g \cdot x = x\} \tag{15}$$

forms a subgroup of $G$.

Here, we give a statement and brief proof of the well-known orbit-stabilizer theorem.

**Theorem A.4** (Orbit-stabilizer theorem). *For an action of a finite group $G$ on a set $X$, for each $x \in X$, the function $\theta_x : G \to X$ defined by*

$$\theta_x(g) := g \cdot x \tag{16}$$

*induces a bijection from the left cosets of $\mathrm{Stab}_G(x)$ to $\mathrm{Orb}_G(x)$. This implies that the orbit $\mathrm{Orb}_G(x)$ is finite and*

$$|\mathrm{Orb}_G(x)| = \frac{|G|}{|\mathrm{Stab}_G(x)|}. \tag{17}$$

*Proof.* We show that $\theta_f$ induces a well defined function on the left-cosets of $\mathrm{Stab}_G(x)$, which we call $\tilde{\theta}_f$. Specifically, we define

$$\tilde{\theta}_f(g \, \mathrm{Stab}_G(x)) := g \cdot x, \tag{18}$$

and show that $\tilde{\theta}_f$ is injective and surjective.

To see that $\tilde{\theta}_f$ is well defined and injective, note that

$$h \in g \, \mathrm{Stab}_G(x) \iff g^{-1}h \in \mathrm{Stab}_G(x) \tag{19}$$
$$\iff g^{-1}h \cdot x = x \tag{20}$$
$$\iff g \cdot x = h \cdot x, \tag{21}$$

using the definition of $\mathrm{Stab}_G$.

For surjectivity, we have

$$y \in \mathrm{Orb}_G(x) \implies \exists g \in G \text{ s.t. } y = g \cdot x \tag{22}$$
$$\implies y = \tilde{\theta}_f(g \, \mathrm{Stab}_G(x)) \tag{23}$$

using the definition of $\mathrm{Orb}_G$. $\qquad \square$

# B Proofs

## B.1 Background (Section 2)

*Proof for Definition 2.2.* We first prove $s \cdot f_\sim \in \mathcal{F}_\sim$ for all $s \in \mathcal{S}_n$ and $f_\sim \in \mathcal{F}_\sim$. Let $s \in \mathcal{S}_n$ and $f, g \in \mathcal{F}$. Now $s \cdot f_\sim \sim s \cdot g_\sim$ exactly if $f \sim g$ because $\sim$ is exchangeable, and the inverse permutation $s^{-1} \in \mathcal{S}_n$ of $s$ exists. Since $\mathcal{F}_\sim$ is a partition of $\mathcal{F}$, this implies $\{s \cdot f \mid f \in f_\sim\} \in \mathcal{F}_\sim$ and therefore $s \cdot f_\sim \in \mathcal{F}_\sim$. It is left to show that the operator $\cdot : \mathcal{S}_n \times \mathcal{F}_\sim \to \mathcal{F}_\sim$ is a left group action. This follows from $\cdot : \mathcal{S}_n \times \mathcal{F} \to \mathcal{F}$ associated with the permutable class $\mathcal{F}$ being a left group action. $\qquad\square$

## B.2 Autoregressive shuffle coding (Section 4)

**Definition B.1** (Exchangeable function)**.** We say a function $a$ on a permutable class $\mathcal{F}$ is *exchangeable* if $a(f) = a(g)$ for all $f, g \in \mathcal{F}$ with $f \simeq g$.

**Lemma B.2.** *If $f \to P(f)$ on $\mathcal{F}$ is exchangeable then the functions $f_{[i]} \to P(f_{[i]})$ on $\mathcal{F}_{[i]}$ for $i \in [n+1]$ are exchangeable.*

*Proof.* If $f \to P(f)$ is exchangeable, then for any $i \in [n+1]$ and $s \in \mathcal{S}_i$, Equation (2) implies $P(s \cdot f_{[i]}) = \sum_{f \in f_{[i]}} P(s \cdot f) = \sum_{f \in f_{[i]}} P(f) = P(f_{[i]})$, and therefore $f \to P(f_{[i]})$ on $\mathcal{F}_{[i]}$ is exchangeable. $\qquad\square$

**Lemma B.3.** *If $f \to P(f)$ on $\mathcal{F}$ is exchangeable, then for $i \in [n], g_i \in \mathcal{F}_{[i]}$, the function $f_{[i]} \to P(g_i \mid f_{[i]})$ on $\mathcal{F}_{[i]}$ is exchangeable, and $P(g_i \mid \overline{f_{[i]}}) = P(g_i \mid f_{[i]})$ for $f_{[i]} \in \overline{f_{[i]}}$.*

*Proof.* Assume $P(f)$ is exchangeable. We will use induction over $i$ starting at $n$ down to $0$, for the statement that $f_{[i]} \to P((g_j)_{j \in [n]\setminus[i]} \mid f_{[i]})$ is exchangeable for any given $g_j$, and implies the lemma. $P((g_j)_{j \in [n]\setminus[n]} \mid f_{[n]}) = P(() \mid f_{[n]})$ is exchangeable, covering the base case. Assume for $i \in [n]$ that $f_{[i+1]} \to P((g_j)_{j \in [n]\setminus[i+1]} \mid f_{[i+1]})$ is exchangeable. For all $i \in [n]$ and $f \in \mathcal{F}$, we have $P(f) = P(f_{[i]})P(f_i \mid f_{[i]})P((f_j)_{j \in [n]\setminus[i+1]} \mid f_{[i+1]})$. Therefore, the function mapping from $f : \mathcal{F}$ to $P(f_i \mid f_{[i]}) = P(f_{[i]})P((f_j)_{j \in [n]\setminus[i+1]} \mid f_{[i+1]})/P(f)$ is also exchangeable due to Lemma B.2 and the inductive assumption. This further implies that the function mapping from $f_{[i]} : \mathcal{F}_{[i]}$ to $P((g_j)_{j \in [n]\setminus[i]} \mid f_{[i]}) = P((g_j)_{j \in [n]\setminus[i+1]} \mid f_{[i+1]})P(g_i \mid f_{[i]})$ is exchangeable, completing the induction. $\qquad\square$

**Lemma B.4.** *For a prefixing chain given by $\mathcal{F}_{[i]}$, $\mathrm{Stab}_{\mathrm{Aut}(f_{[i+1]})}(i) = \mathrm{Aut}(f_{[i]})$ for all $i \in [n-1], f \in \mathcal{F}$.*

*Proof.* We have $\mathrm{Stab}_{\mathrm{Aut}(f_{[i+1]})}(i) = \{s \in \mathcal{S}_{i+1} \mid s \cdot i = i \wedge s \cdot f \sim_{[i+1]} f\}$, which is $= \{s \in \mathcal{S}_i \mid s \cdot f \sim_{[i+1]} f\}$ due to Equation (1) and $= \{s \in \mathcal{S}_i \mid s \cdot f \sim_{[i]} f \wedge \bigcup (s \cdot f)'_{[i+1]} = \bigcup \overline{f'_{[i+1]}}\}$ through Equation (7). Since $(s \cdot f)_{[i+1]}$ is isomorphic to $f$ in $\mathcal{F}'_{[i+1]}$, this is in turn $= \{s \in \mathcal{S}_i \mid s \cdot f \sim_{[i]} f\} = \mathrm{Aut}(f_{[i]})$. $\qquad\square$

*Proof of Equation (8).* The optimal rate is given by Equation (3):

$$\log \frac{1}{P(\overline{f_{[i]}})} = \underbrace{\log \frac{1}{P(f_{[i]})}}_{\text{Ordered rate}} - \underbrace{\log \frac{i!}{|\mathrm{Aut}(f_{[i]})|}}_{\text{Discount}}. \tag{24}$$

The optimal rate for $\overline{g_{[i-1]}}$ is given by Equation (24):

$$\log \frac{1}{P(\overline{g_{[i-1]}})} = \underbrace{\log \frac{1}{P(g_{[i-1]})}}_{\text{Ordered rate}} - \underbrace{\log \frac{(i-1)!}{|\mathrm{Aut}(g_{[i-1]})|}}_{\text{Discount}}. \tag{25}$$

It is informative to rewrite Equation (24) in terms of Equation (25):

$$\log \frac{1}{P(\overline{f_{[i]}})} = \log \frac{1}{P(\overline{g_{[i-1]}})} + \log \frac{P(g_{[i-1]})}{P(f_{[i]})}$$
$$- \log \frac{i \cdot |\mathrm{Aut}(g_{[i-1]})|}{|\mathrm{Aut}(f_{[i]})|}. \tag{26}$$

Definition 4.2 and Lemma B.3 imply $P(f_{[i]}) = P((s \cdot f)_i) = P(g_{[i]}) = P(g_{i-1} \mid \overline{g_{[i-1]}}) \cdot P(g_{[i-1]})$.

Additionally, we have $|\mathrm{Aut}(f_{[i]})| = |\mathrm{Aut}\left(s \cdot f_{[i]}\right)| = |\mathrm{Aut}(g_{[i]})|$ since permuting $f$ results in a conjugated automorphism group which is isomorphic to $\mathrm{Aut}(f_{[i]})$. Lemma B.4 together with the orbit-stabilizer theorem A.4 implies $|\mathrm{Aut}(g_{[i]})| = |\mathrm{Orb}_{\mathrm{Aut}(g_{[i]})}(i-1)| \cdot |\mathrm{Aut}(g_{[i-1]})|$. Finally, we have $|\mathrm{Orb}_{\mathrm{Aut}(g_{[i]})}(i-1)| = |\mathrm{Orb}_{\mathrm{Aut}(f_{[i]})}(j)|$. We can now rewrite Equation (26) into Equation (8). □

**Lemma B.5.** UnorderedPrefix *forms a valid codec, with* encode *and* decode *being inverses of each other.*

*Proof.* This is mostly straightforward, but notably, the swap operation of the encoder is not reversed. Here, we exploited that output of the function $h_{[i-1]} \to \mathrm{orbits}_i(\mathrm{push}(h_{[i-1]}, g_{i-1}))_{i-1}$ on $\mathcal{F}_{[i+1]}$ does not change when permuting the input, and therefore the decoder recovers the correct orbit index $\mathsf{o} = \mathrm{orbits}_i(g_{[i]})_{i-1} = \mathrm{orbits}_i(f_{[i]})_j$ for any decoded $h_{[i-1]} \in \overline{g_{[i-1]}}$. □

## B.3 Incomplete shuffle coding (Section 3)

*Proof for Equation (4).* The condition in Definition 3.1 is equivalent to $f \simeq g$ for all $f, g \in \mathcal{F}$ with $f \sim g$. We show $\bar{f} = \bigcup \overline{f_\sim}$ by proving that for $g_\sim \in \mathcal{F}/\sim$, we have $g_\sim \in \overline{f_\sim}$ exactly if $g_\sim \subseteq \bar{f}$. $g_\sim \in \overline{f_\sim}$ means $\exists s \in \mathcal{S}_n : g_\sim = s \cdot f_\sim$, which is equivalent to $\exists s \in \mathcal{S}_n : g \sim s \cdot f$. On the other hand, $g_\sim \subseteq \bar{f}$ holds exactly if $g_\sim \subseteq \{t \cdot f_\sim \mid t \in \mathcal{S}_n\}$, meaning that $\exists t \in \mathcal{S}_n : g = t \cdot f$. It is left to show that $\exists s \in \mathcal{S}_n : g \sim s \cdot f$ exactly if $\exists t \in \mathcal{S}_n : g = t \cdot f$. The implication $\Leftarrow$ is proved with $s := t$. To show the implication $\Rightarrow$, assume $\exists s \in \mathcal{S}_n : g \sim s \cdot f$. Definition 3.1 now implies $\exists s \in \mathcal{S}_n : g \simeq s \cdot f$, and therefore $\exists s, u \in \mathcal{S}_n : g = u \cdot s \cdot f$. $t := u \cdot s$ implies $\exists t \in \mathcal{S}_n : g = t \cdot f$, completing the proof.

This leads to $P(\bar{f}) = P(\overline{f_\sim})$ for any given distribution $P(f)$ over $\mathcal{F}$, and Equation (4) follows from applying Equation (3) on both sides. □

*Proof for Equation (5).* To achieve this runtime, we implicitly assumed that our ordered graph model is fast enough. We can hash a multiset represented as a string in linear time by accumulating hashes using a bitwise 'xor' operation, which is associative and commutative, making the overall hash permutation-invariant. This results in an overall runtime of $O(\sum_{i \in [n]} |n_i(g)|) = O(m)$. The remaining computation has the runtime $O(n \log n)$ of joint shuffle coding on multisets, leading to the given overall runtime complexity. □

## C Color refinement details

We here state the formal definition of color refinement, as described in Section 3, that we use for the paper:

**Definition C.1.** For an (ordered) simple graph $g \in \mathcal{G}_n$, let $n_i(g)$ denote the set of indices of the neighbors of the vertex at index $i$. Let the *coloring* of a graph $g \in \mathcal{G}_n$ with $k \in \mathbb{N}$ iterations be the tuple $C_k(g)$ with $C_0(g) = (|n_i(g)|)_{i \in [n]}$ and

$$C_{k+1}(g) = (h(\{\!\{C_k(g)_j \mid j \in n_i(g)\}\!\}))_{i \in [n]}, \tag{27}$$

for a 'hashing' function $h : \mathcal{M} \to \mathbb{N}$ on multisets $\mathcal{M} = \cup_{i \in [n+1]} \overline{\mathbb{N}^i}$ of up to $n$ natural numbers $\mathbb{N}$. We then obtain a congruence corresponding to $C_k(f) = C_k(g)$ and $\mathcal{G}_n / \sim_{C_k}$ forms a class of incompletely ordered objects for $k \in \mathbb{N}$.

Color refinement is often run to convergence of vertex partitions (Huang and Villar, 2021). This happens after at most $[n]$ iterations and is therefore equivalent to $C_n$. It can be naturally extended to graphs with vertex and edge attributes, by hashing each vertex attribute together with the multiset of neighboring edge attributes in the initial $C_0$, and then convolving multisets of pairs of neighboring edges attributes and hashes in subsequent iterations. The corresponding Weisfeiler Leman graph isomorphism test is based on the fact that $\overline{C_k(f)} \neq \overline{C_k(g)}$ implies $f \not\simeq g$ and successfully distinguishes almost all pairs of non-isomorphic graphs (Babai and Kučera, 1979).

Similarly, color refinement with $k > 0$ convolutions finds the exact automorphism group $\mathrm{Aut}(g_{\sim_{C_k}}) = \mathrm{Aut}(g)$ for almost all simple graphs for large enough $n$.

*Proof.* Babai et al. (1980) show that with probability of at least $1 - \frac{1}{n^7}$, there is a set $U$ of vertices whose degrees are distinct in $U$, and no other vertices have the same set of neighbors within $U$. Since color refinement starts with the vertex degrees, a single convolution will assign pairwise distinct hashes for almost all graphs. $\square$

We assumed here that no hash collisions occur ($h(a) = h(b)$ with $a \neq b$), for simplicity. In practice, we use an off-the-shelf 64-bit hash function $h$, where hash collisions are extremely improbable. Even if such a collision occurs, it results only in a marginal rate increase.

# D Conceptual visualizations

We visualize key concept for autoregressive shuffle coding for the example of graphs in Table 4, and summarize the variants of shuffle coding proposed in Table 5.

Table 4: Examples of key concepts from Section 4 for the permutable class $\mathcal{F}$ of simple graphs with 5 vertices and the prefixing chain given by Example 4.2. Dotted edges indicate deleted information.

| Concept | Example |
| --- | --- |
| Unordered derivative | $\overline{\left(\begin{smallmatrix}0 & 1 \\ 4 & 2 \\ & 3\end{smallmatrix}\right)}' = \overline{\left(\begin{smallmatrix}\cdot & \cdot \\ 4 & 2 \\ & \cdot\end{smallmatrix}\right)} \in \overline{\mathcal{F}'}$ |
| Unordered $i$-th derivative | $\overline{\left(\begin{smallmatrix}0 & 1 \\ 4 & 2 \\ & 3\end{smallmatrix}\right)}^{(3)} = \overline{\left(\begin{smallmatrix}\cdot & \cdot \\ 4 & 2 \\ & 3\end{smallmatrix}\right)} = \left\{ \begin{smallmatrix}0 & 1 \\ 4 & 2 \\ & 3\end{smallmatrix}, \begin{smallmatrix}0 & 1 \\ 4 & 2 \\ & 3\end{smallmatrix} \right\} \in \overline{\mathcal{F}^{(3)}}$ |
| Prefix | $\left(\begin{smallmatrix}0 & 1 \\ 4 & 2 \\ & 3\end{smallmatrix}\right)_{[3]} = \begin{smallmatrix}0 & 1 \\ 4 & 2 \\ & 3\end{smallmatrix} = \left\{ \begin{smallmatrix}0 & 1 \\ 4 & 2 \\ & 3\end{smallmatrix}, \begin{smallmatrix}0 & 1 \\ 4 & 2 \\ & 3\end{smallmatrix} \right\} \in \mathcal{F}_{[2]}$ |
| Slice | $\left(\begin{smallmatrix}0 & 1 \\ 4 & 2 \\ & 3\end{smallmatrix}\right)_2 = \begin{smallmatrix}\cdot & \cdot \\ 4 & 2 \\ & 3\end{smallmatrix}$ , represented by $\begin{smallmatrix}4 & - & 2 \\ & 3 &\end{smallmatrix}$ in context of $\begin{smallmatrix}\cdot & \cdot \\ 4 & 2 \\ & 3\end{smallmatrix}$ |
| Pop | $\mathrm{pop}\left(\begin{smallmatrix}0 & 1 \\ 4 & 2 \\ & 3\end{smallmatrix}\right) = \left(\begin{smallmatrix}0 & 1 \\ 4 & 2 \\ & 3\end{smallmatrix}, \begin{smallmatrix}\cdot & \cdot \\ 4 & 2 \\ & 3\end{smallmatrix}\right)$ |
| Push | $\mathrm{push}\left(\left(\begin{smallmatrix}0 & 1 \\ 4 & 2 \\ & 3\end{smallmatrix}, \begin{smallmatrix}\cdot & \cdot \\ 4 & 2 \\ & 3\end{smallmatrix}\right)\right) = \begin{smallmatrix}0 & 1 \\ 4 & 2 \\ & 3\end{smallmatrix}$ |
| Orbit of index $i$ | $\mathrm{Orb}_{\mathrm{Aut}\left(\begin{smallmatrix}0 & 1 \\ 4 & 2 \\ & 3\end{smallmatrix}\right)}(1) = \{1, 3\}$ |
| Orbits | $\mathrm{Orbs}_{\mathrm{Aut}\left(\begin{smallmatrix}0 & 1 \\ 4 & 2 \\ & 3\end{smallmatrix}\right)} = \{\{0\}, \{1, 3\}, \{2, 4\}\}$ |
| Orbit function | $\mathrm{orbits}\left(\begin{smallmatrix}0 & 1 \\ 4 & 2 \\ & 3\end{smallmatrix}\right) = (2, 0, 1, 0, 1)$ implies $\mathrm{orbits}\left(\begin{smallmatrix}0 & 1 \\ 4 & 2 \\ & 3\end{smallmatrix}\right) = (0, 0, 1, 2, 1)$ |

Table 5: Comparison between variants of shuffle coding.

| Shuffle Coding Variant | Not One-Shot & Allows Any Model | One-Shot & Requires Autoregressive Model |
|---|---|---|
| Optimal Rate & Requires Automorphism Group | **Complete Joint** (Kunze et al., 2024) | **Complete Autoregressive** |
| Near-Optimal Rate & Allows Approx. Automorphism Group | **Incomplete Joint** | **Incomplete Autoregressive** |

# E    Chunking

| $C$ | no chunking ('full') $[1, 1, 1, 1, 1]$ | $[2, 2, 1]$ | single chunk ('joint') $[5]$ |
|---|---|---|---|
| $f_{[1]}$ | a | ab | abacb |
| $f_{[2]}$ | ab | ab | abacb |
| $f_{[3]}$ | aba | abac | abacb |
| $f_{[4]}$ | abac | abac | abacb |
| $f_{[5]}$ | abacb | abacb | abacb |

Table 6: Prefixes for the string $f = \text{abacb}$ according to chunked prefixing chains of Example 4.1, for various chunk size sequences $C$. When used with autoregressive shuffle coding, the prefix $f_{[i]}$ resembles the information decoded after $i$ iterations during decoding.

Autoregressive shuffle coding with chunking, as described in Section 4.3 refers to autoregressive shuffle coding using the following 'chunked' prefixing chain:

**Definition E.1** (Chunked prefixing chain). Let $(\mathcal{F}_{[i]})_{i \in [n]}$ be a prefixing chain on $\mathcal{F}$ with $\mathcal{F}_{[i]} := \mathcal{F}^{(n-i)}/\sim_i$ and let $C = (c_j)_{j \in [c]}$ be a sequence of $c$ chunk sizes with $\sum_{j \in [c]} c_j = n$. In this context, we refer to $p_j := \sum_{k \in [j+1]} c_k$ as the prefix size for chunk $j \in [c]$, and $C(i) := \min\{j \in [c] | i \in [p_j]\}$ as chunk for index $i \in [n]$. Another prefixing chain is now formed on $\mathcal{F}$ by $(\mathcal{F}^{(n-i)}/\sim_{p_{C(i)}})_{i \in [n]}$, which we refer to as the *chunked prefixing chain* of $(\mathcal{F}_{[i]})_{i \in [n]}$ with chunk sizes $C$.

Then, the slice at the first position of a chunk codes a complete chunk from the original prefixing chain, and all its remaining slices have no additional information about the object. To apply incomplete autoregressive shuffle coding according to some incompletely ordered prefixes $(f_{[i]})_\sim \in \mathcal{F}_{[i]}/\sim$, formally, we will require a prefixing chain to be defined for each $\mathcal{F}_{[i]}/\sim$. Then, we code orbits of chunked prefixes of length $i$ according to $\text{Aut}(((f_{[p_{C(i)}]})_\sim)_{[i]})$. This means that we approximate the orbit based on the full chunked prefix, and then use a prefix of that approximation of length $i$.

## E.1    Runtime

For graphs, we will use color refinement to map to incompletely ordered prefixes $(f_{[i]})_{\sim_{C_k}}$ and use the string prefixes according to Example 4.1 of the resulting hashes within a given chunk. This choice requires $c$ color refinement runs (instead of one as in Equation (5)), and additionally has the runtime complexity of (full) autoregressive shuffle coding on strings $O(n \log n)$, resulting in the overall runtime $O(c \cdot m + n \log n)$ stated in Equation (10), which achieves the goal of a practical runtime for a fixed number of chunks $c$. Although not affecting runtime complexity, we can also switch adaptive entropy coding to a Fenwick tree to achieve better memory layout and therefore runtime, as described in Section 4.3, since all possible vertex colors within a chunk are known immediately after color refinement.

### E.2   Choosing chunk sizes

Larger chunks soften the restriction for our ordered model to be autoregressive but increase the (one-shot) compression rate due to more required initial bits. For chunking on graphs as discussed above, larger chunks additionally can expect a speedup due to fewer rehashings with color refinement, and a slightly worse net rate, since chunk borders soften the restriction of the automorphism group to be factored.

The degenerate case of $c = n$ chunks is equivalent to not using chunking at all, a setting we refer to as 'full' autoregressive shuffle coding, shown in the second column of Table 6. The last column shows the other extreme with only $c = 1$ chunk, where autoregressive shuffle coding degenerates into a method behaving very much like joint shuffle coding, in the sense that we can employ a joint model, and the method is not suitable for one-shot coding since the complete discount would stay unrealized in the rate due to initial bits. We therefore refer to this setting as 'joint' autoregressive shuffle coding. Joint shuffle coding still has a reason to exist: It typically allows more batched and parallel coding of order information and accordingly, it can be significantly faster, as we show in our experiments.

The unrealized discount in the rate due to initial bits is influenced by the distribution of the object's information among slices, as visualized in Figure 2. With the usual prefixing chain from Example 4.1 for i.i.d. multisets, information is distributed equally across slices, as pictured in Figure 2(a), and all but $O(\log n)$ of the $O(n \log n)$ discount should be realized in the rate. For simple graphs sampled from an Erdős-Rényi model and the prefixing chain from Example 4.2, each slice $f_i$ contains information about $n - 1 - i$ edges, so the expected slice rate drops linearly with $i$ down to 0 for the last slice, as visualized in Figure 2(b). In effect, the last slices (that are encoded onto the stack first) have the least amount of information, canceling out fewer initial bits. For graphs, we can thus expect less of the discount realized in the rate.

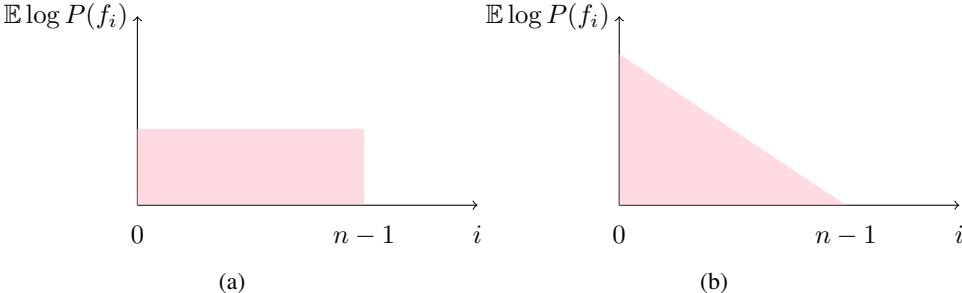

Figure 2: Expected slice information $\mathbb{E} \log P(f_i)$ by slice index $i$ for (a) i.i.d. strings using the prefixing chain from Example 4.1 and (b) simple Erdős-Rényi graphs using Example 4.2.

Assuming that all slices have the same rate, which is met in expectation for i.i.d. strings as shown in Figure 2(a), that each coded orbit has the same rate (which is only slightly inaccurate), and given that we want to use exactly $c$ chunks, the chunk sizes that minimize the rate have the following property: For each pair of neighboring chunks with sizes $c_i$ and $c_{i+1}$, the ratio of the two latter by the form size should be approximately equal to the *relative discount*, meaning the discount $\log \frac{n!}{\mathrm{Aut}(f)}$ of the object $f$ per ordered net rate $\log P(f)$,

$$\frac{c_{i+1}}{c_i} \approx \frac{\log \frac{n!}{\mathrm{Aut}(f)}}{\log P(f)}, \tag{28}$$

leading to a geometric series of chunk sizes with the given base, with the first chunk being the largest.[5] For simple Erdős-Rényi graphs, as shown in Figure 2(b), the slice rate is not uniform, making it more difficult to model the condition required for the minimal rate. For our experiments on graphs, we will simply choose (approximately) equally sized chunks, and leave optimizing relative chunk sizes for future work.

---

[5] During encoding, this condition causes the ANS stack to be exactly emptied by the initial bits of the next chunk. It can be shown that this is optimal under the given assumptions.

# F   Autoregressive Pólya urn model details

Kunze et al. (2024) use the Pólya urn (PU) preferential attachment model proposed by Severo et al. (2023b) to drastically improve compression rate on SZIP graphs with joint shuffle coding. Motivated by this, we use a variational approximation to obtain a tractable autoregressive model that we refer to as autoregressive Pólya urn (AP). The task of predicting the next graph slice $f_i$ given the prefix $f_{[i]}$ can be broken down into two parts: Given the prefix $f_{[i]}$, predict the number of edges $k_i \in [n-i]$ within the next slice $f_i$, and then from that, predict the edge positions within the slice. For the Pólya urn (PU) model $P(f)$, assuming the slice representation from Example 4.2, the distribution for the first part, $P(k_i|f_{[i]})$, is not tractable, and in order to obtain a tractable variational autoregressive model, we instead approximate it based on a Zipf distribution. Specifically, we use the Zipf distribution $Q(k_i) \propto \frac{1}{(k_i+1)^2}$, except for smaller graphs up to 100000 edges (which includes all SZIP graphs), for which we use $Q(k_i) \propto \frac{1}{k_i^2}$ for $k_i > 1$ and $Q(k_i = 0) = Q(k_i = 1) \approx \frac{1}{3}$. The distribution of the second part, $P(f_i|f_{[i]}, k_i)$, is tractable. The generative process for $P(f_i|f_{[i]}, k_i)$ is as following: Given the graph of the prefix $f_{[i]}$ and the number of edges $k_i$ in the next slice $f_i$, iteratively insert $k_i$ edges, with probabilities proportional to the neighbor count $+1$ of the adjacent vertices $\{i+1, i+2, \ldots n-1\}$ without allowing repetition (i.e. zeroing out the probabilities where edges were already inserted). Our overall autoregressive model is then given by $Q(f_i \mid f_{[i]}) = \sum_{k_i=0}^{i} P(f_i \mid f_{[i]}, k_i)Q(k_i \mid f_{[i]})$.

In general, it is not important for an autoregressive to exactly match or even approximate recognizable joint models. Instead, they can be designed or learned on their own. In our case, however, a joint model inspired a reasonable autoregressive model through approximating the original with the derivation above.

# G  Multiset compression results

All shuffle coding speeds in this paper were measured on a MacBook Pro 2018 with a 2.7GHz Intel Core i7 CPU.

We compare our implementation of full autoregressive shuffle coding to Severo et al. (2023a) on multisets of varying lengths, sampled from an i.i.d. categorical distribution with probabilities sampled according to a Dirichlet distribution, as described in Severo et al. (2023a). We show compression rates in Table 7, and speeds in Figure 3. The results confirm that our implementation matches the rate of Severo et al. (2023a), but is well over two orders of magnitude faster, and scales to large multisets. We observe that the rate is within 64 bits of the optimal rate across all multiset sizes, likely mostly caused by the fixed overhead of ANS. This is consistent with our prediction from Appendix E.2 that the unrealized discount should be very small at $O(\log n)$.

We also evaluate joint shuffle coding using the same data and i.i.d. model, as well as 'joint' autoregressive shuffle coding (with a single chunk). While both result in the same net rate as full autoregressive shuffle coding, joint shuffle coding leads to the fastest runtime over a wide range of multiset sizes.

In Table 7, we also report the rates of autoregressive shuffle coding with 10 chunks using a geometric series of chunk sizes with varying choices for the base, compared to equally-sized chunks (base 1). The results confirm the prediction of Appendix E.2 that on multisets, a geometric series of (unequal) chunk sizes can lead to better rates over equally sized chunks. For our experiments, our chunked autoregressive ordered model simply codes multiple slices per chunk using the underlying fully autoregressive model. On multisets, this results in autoregressive shuffle coding with larger chunks having no benefits over the fully autoregressive variant. Improved speeds can be expected for specialized chunked autoregressive models that are parallelized, similar to joint shuffle coding. We leave this for future work.

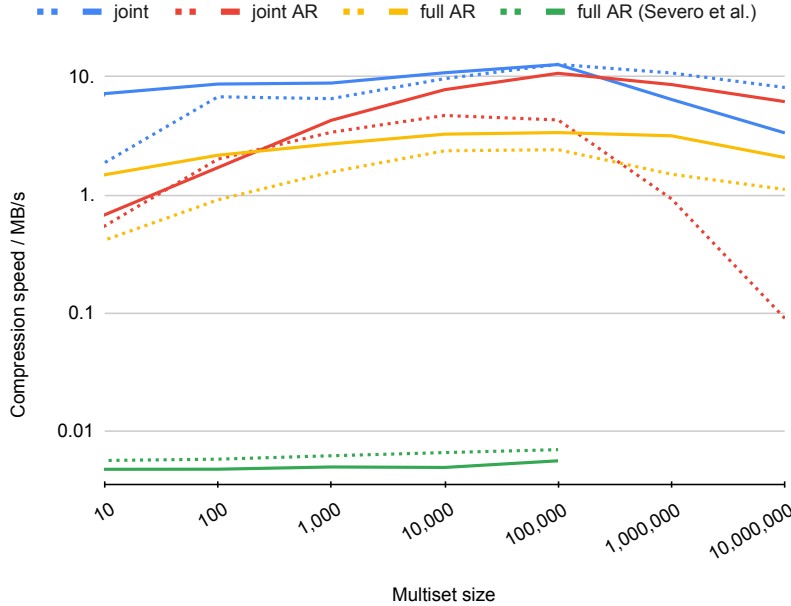

Figure 3: Compression speeds (dotted lines) and decompression speeds (solid lines) of multisets of varying size using joint and autoregressive shuffle coding with 1 chunk ('joint AR') and $n$ chunks ('full AR'), compared to the full autoregressive implementation from Severo et al. (2023a). All results are based on the (ordered) string rate as the reference uncompressed size averaged across 10 runs (100 runs for sizes <1M for our implementations).

Table 7: Compression rates in bits for one-shot compression of multisets with full autoregressive shuffle coding, compared to the implementation from Severo et al. (2023a), as well as the optimal (net) rate. We also show the effect rates when using 10 chunks with a geometric series of sizes, with varying bases. The last but one column uses the relative discount from Equation (28) of each graph as the base (shown in parentheses).

| | Full | | | 10 Chunks (Geometric with given Base) | | | |
| --- | --- | --- | --- | --- | --- | --- | --- |
| Size | Net | Ours | Severo | 1 (unif.) | $(\log \frac{n!}{\mathrm{Aut}(f)})/\log P(f)$ | | 93% |
| 1k | 1769 | 1833 | 1856 | 2707 | 2487 | (18%) | 2443 |
| 10k | 8683 | 8744 | 8768 | 17598 | 15219 | (49%) | 14988 |
| 100k | 68852 | 68912 | 68960 | 157690 | 131782 | (82%) | 131782 |
| 1M | 663046 | 663104 | – | 1550754 | 1295534 | (91%) | 1295534 |
| 10M | 6597373 | 6597432 | – | 15473751 | 12935710 | (93%) | 12935710 |

# H   Joint graph shuffle coding results

We report experimental results for incomplete joint shuffle coding on graphs in Figure 4 and Tables 8 to 10.

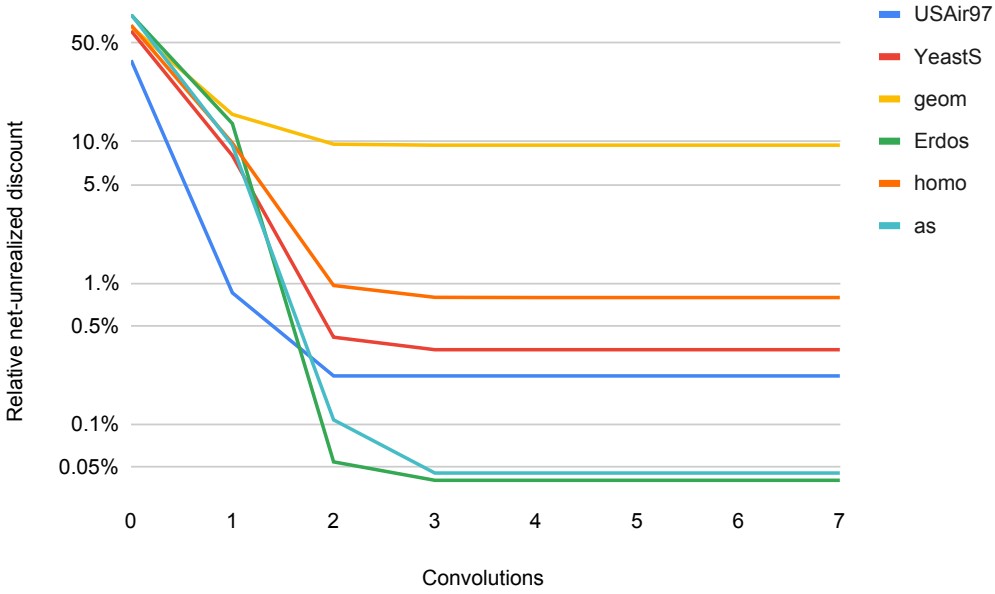

Figure 4: Rate increase from Equation (4) relative to the (optimal) discount from Equation (3) for incomplete shuffle coding on graphs, depending on the number of color refinement convolutions $k$, for each SZIP graph. The results are independent of the employed ordered model.

Table 8: Incomplete joint shuffle coding with color refinement using 3 convolutions on the TUDatasets (Morris et al., 2020), compared to complete joint shuffle coding from Kunze et al. (2024), both using the Erdős-Rényi (ER) model. Since complete joint shuffle coding is too slow for three of the 24 social network datasets, REDDIT-BINARY, REDDIT-MULTI-5K, REDDIT-MULTI-12K, they were evaluated separately in the category 'Reddit'. Compression rates are measured in bits per edge, and encoding and decoding speeds are for a single thread, in kB/s.

| | | | | Speed (1 Thread) | | | |
| | Rate | | | Complete | | Incompl. | |
| Graph type | Compl. | Incompl. | Increase | Enc. | Dec. | Enc. | Dec. |
|---|---|---|---|---|---|---|---|
| Small molecules | 1.14 | 1.19 | 4.0% | 54 | 56 | 167 | 179 |
| Bioinformatics | 6.50 | 6.51 | 0.1% | 51 | 66 | 426 | 421 |
| Computer vision | 4.49 | 4.56 | 1.5% | 25 | 28 | 163 | 162 |
| Social networks | 2.97 | 2.97 | 0.3% | 0.44 | 0.47 | 215 | 284 |
| Synthetic | 2.99 | 2.99 | 0.1% | 98 | 110 | 317 | 323 |
| Reddit | – | 5.38 | – | – | – | 129 | 123 |

Table 9: Incomplete joint shuffle coding with color refinement using 3 convolutions on the SZIP dataset, compared to complete joint shuffle coding from Kunze et al. (2024), both using the autoregressive Pólya urn (AP) model. We report the *net* compression rate, that is the additional cost of compressing that graph assuming there is already some compressed data to append to, measured in bits per edge, as well as compression and decompression speeds on 8 threads in kB/s. All results are means across multiple runs, 3 for complete, and 100 for incomplete shuffle coding.

| | | | | Speed (8 Threads) | | | |
| | Net rate | | | Complete | | Incompl. | |
| SZIP graph | Compl. | Incompl. | Incr. | Enc. | Dec. | Enc. | Dec. |
|---|---|---|---|---|---|---|---|
| Airports (USAir97) | 3.12 | 3.13 | 0.2% | 98 | 111 | 1025 | 1151 |
| Protein (YeastS) | 5.89 | 5.91 | 0.2% | 2.834 | 2.852 | 1636 | 1697 |
| Collaboration (geom) | 4.57 | 4.84 | 7.1% | 0.004 | 0.004 | 2289 | 2372 |
| Collaboration (Erdos) | 4.45 | 4.45 | 0.2% | 0.024 | 0.023 | 1951 | 1979 |
| Genetic (homo) | 6.95 | 6.98 | 0.5% | 0.249 | 0.261 | 2238 | 2358 |
| Internet (as) | 4.52 | 4.53 | 0.2% | 0.002 | 0.003 | 2459 | 2537 |

# I   Autoregressive graph shuffle coding results

To evaluate the effect of chunk size, we run autoregressive shuffle coding with varying numbers of equally sized chunks on the SZIP dataset. We report discount unrealized by the net rate, which is independent of the ordered model[6], as well as results on the AP model in Figure 5. The results confirm that finer chunks lead to better one-shot rates, with most of the discount realized in the rate for roughly 16 chunks or more, with practical runtimes. We therefore use a default of 16 equally-sized chunks in further experiments.

The results also confirm that finer chunks lead to slightly better net rates, an effect predicted in Appendix E.2 caused by the corresponding softening of the restriction of a factored approximate automorphism group retrieved through color refinement. Remarkably, we observe that the net rate of autoregressive shuffle coding with 512 chunks is only 2 bits above the optimal rate given by Equation (3) for the SZIP graph dataset in total, leaving only 0.00002% of the discount unrealized in the net rate, practically matching the optimal net rate from Equation (3), and outperforming incomplete joint shuffle coding. For a more practical 16 chunks, 0.05% of the discount is unrealized in the net rate.

Further results, discussed in Section 6, are shown in Tables 11 to 13.

---

[6]It has been evaluated based on results from the deterministic ER codec to avoid stochasticity.

Table 10: Incomplete joint shuffle coding with the autoregressive Pólya urn model (AP), compared to just ordered AP, on SZIP and REC graphs. We report mean net rates and compression speeds based on 10 compression runs (100 for SZIP graphs) with varying initial message seeds. All net rates are in bits per edge, with empirical standard deviations below 0.02 bits per edge except where shown. Single-threaded and multi-threaded compression speeds with 8 logical cores are reported, all in MB/s.

| | | | | | | | | |
|---|---|---|---|---|---|---|---|---|
| | | | | | | Speed | | |
| | | | Net rate | | 1 Thread | | 8 Threads | |
| Graph | Vertices | Edges | Ord. | Incompl. | Enc. | Dec. | Enc. | Dec. |
| **SZIP** | | | | | | | | |
| USAir97 | 0.3k | 2.1k | 4.17 | 3.13 | 1.2 | 1.3 | 1.0 | 1.2 |
| YeastS | 2.3k | 6.6k | 9.17 | 5.91 | 1.6 | 1.6 | 1.6 | 1.7 |
| geom | 6.2k | 21.5k | 7.51±.10 | 4.84±.07 | 2.0 | 2.1 | 2.3 | 2.4 |
| Erdos | 6.9k | 11.9k | 9.88 | 4.45 | 1.8 | 1.8 | 2.0 | 2.0 |
| homo | 8.6k | 26.1k | 10.65 | 6.98 | 1.1 | 1.2 | 2.2 | 2.4 |
| as | 25.9k | 52.4k | 10.24 | 4.53 | 2.1 | 2.2 | 2.5 | 2.5 |
| **REC** | | | | | | | | |
| YouTube | 3.2M | 9.3M | 15.34 | 8.97 | 1.7 | 1.5 | 2.3 | 1.9 |
| FourSquare | 0.6M | 3.2M | 9.95 | 6.61 | 1.6 | 1.5 | 2.1 | 2.0 |
| Digg | 0.8M | 5.9M | 10.69 | 8.56 | 1.8 | 1.7 | 2.4 | 2.2 |
| Gowalla | 0.2M | 1.0M | 13.09 | 9.82 | 2.1 | 2.1 | 2.7 | 2.6 |
| Skitter | 1.6M | 11.1M | 14.44 | 11.59 | 2.1 | 1.9 | 2.9 | 2.5 |
| DBLP | 0.2M | 1.0M | 16.21 | 11.24 | 2.3 | 2.3 | 2.8 | 2.7 |

Table 11: Compression rates and speeds between incomplete autoregressive shuffle coding with 16 (or 200) chunks and 3 color refinement convolutions using an Erdős-Rényi (ER) and our autoregressive Pólya urn (AP) model, compared to the best results obtained by SZIP (Choi and Szpankowski, 2012) for each graph (on different hardware). We report means and standard deviations based on 100 compression runs with varying initial message seeds. All rates are in bits per edge, and all speeds are for 8 threads, in kB/s.

| | | | | | | | |
|---|---|---|---|---|---|---|---|
| | Incomplete Autoregressive Shuffle Coding | | | | | | |
| Rate | ER | | AP | | AP, $c = 200$ | | SZIP |
| Airports (USAir97) | 5.27±.02 | | 3.32±.03 | | **3.27**±.03 | | 3.81 |
| Protein (YeastS) | 7.41±.04 | | 6.46±.04 | | **6.27**±.04 | | 7.05 |
| Collaboration (geom) | 8.73±.09 | | **5.24**±.19 | | **5.10**±.21 | | **5.28** |
| Collaboration (Erdos) | 8.05±.08 | | 5.55±.08 | | 5.24±.08 | | **5.08** |
| Genetic (homo) | 8.81±.03 | | 7.55±.02 | | **7.32**±.03 | | 8.49 |
| Internet (as) | 9.42±.11 | | 5.75±.08 | | **5.46**±.08 | | 5.75 |
| Speed (8 Thread) | Enc. | Dec. | Enc. | Dec. | Enc. | Dec. | Enc. |
| Airports (USAir97) | 301 | 390 | 282 | 433 | 29 | 53 | (164) |
| Protein (YeastS) | 145 | 147 | 576 | 795 | 56 | 89 | (77) |
| Collaboration (geom) | 93 | 86 | 916 | 1253 | 113 | 201 | (64) |
| Collaboration (Erdos) | 46 | 42 | 724 | 983 | 81 | 123 | (18) |
| Genetic (homo) | 63 | 58 | 854 | 1140 | 119 | 215 | (32) |
| Internet (as) | 17 | 15 | 1025 | 1408 | 152 | 277 | (7) |

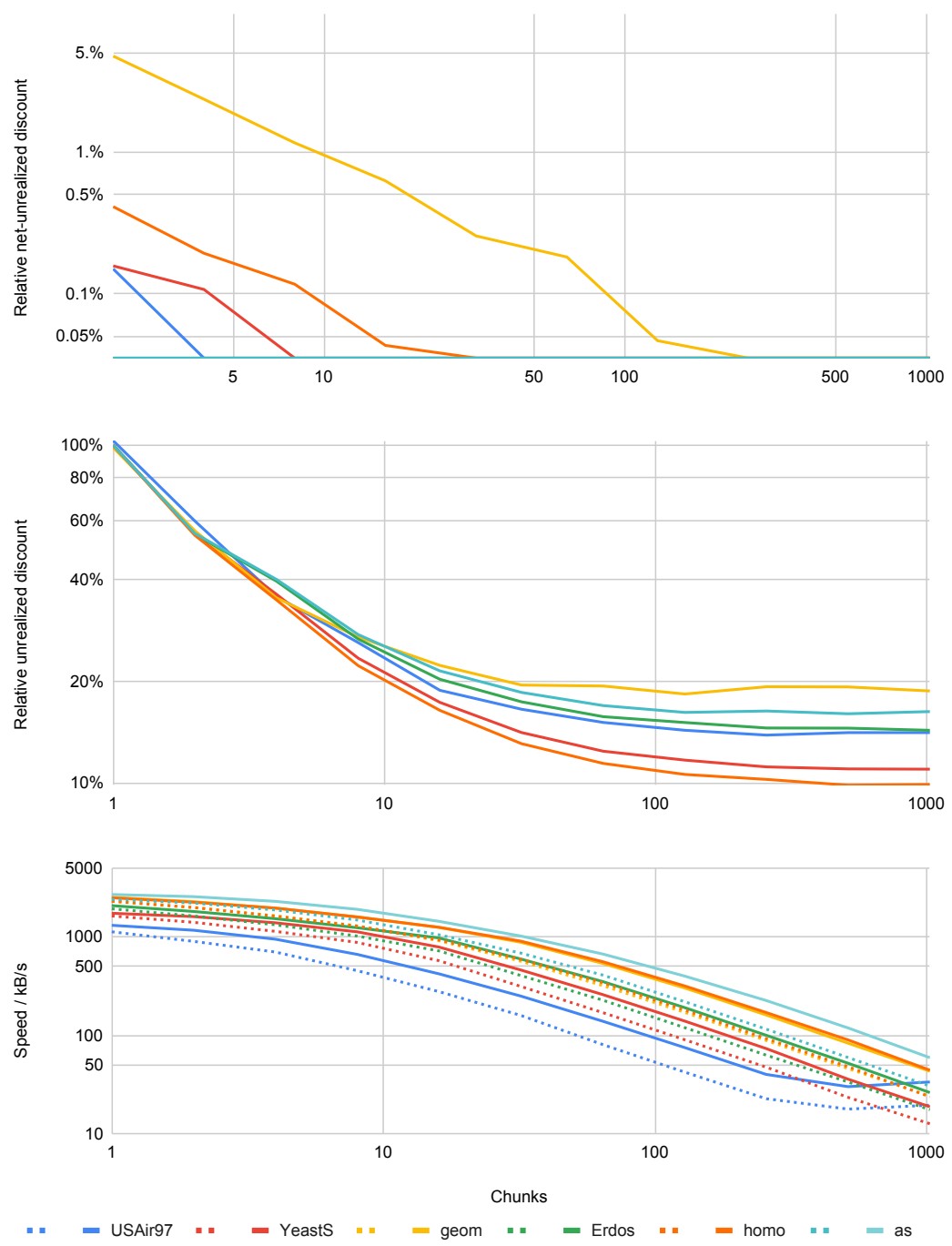

Figure 5: Results for incomplete autoregressive shuffle coding for various numbers of chunks on SZIP graphs, based on the AP model and color refinement with 3 convolutions, averaged across 100 repeated runs per data point. The top and middle plots respectively show the increase of the net rate and rate over the optimal rate, relative to the discount given by Equation (3). The bottom plots shows compression speeds (dotted lines) and decompression speeds (solid lines), based on the (ordered) Erdős-Rényi rate as the reference uncompressed size.

Table 12: Compression rates and speeds of incomplete autoregressive shuffle coding with 16 chunks and 3 color refinement convolutions based on the AP model, for the REC graph dataset. All rates are reported in bits per edge. We show the ordered rate for AP for comparison. Single-threaded and multi-threaded compression speeds are reported in MB/s.

| | | | Ord. | | | Speed | | | |
| | | | | | | 1 Thread | | 8 Threads | |
| Graph | Vertices | Edges | Rate | Rate | Net rate | Enc. | Dec. | Enc. | Dec. |
|---|---|---|---|---|---|---|---|---|---|
| YouTube | 3.2M | 9.3M | 15.34 | 10.05 | 8.93 | 0.5 | 0.7 | 0.9 | 1.3 |
| FourSquare | 0.6M | 3.2M | 9.95 | 7.12 | 6.61 | 0.6 | 1.0 | 1.0 | 1.6 |
| Digg | 0.8M | 5.9M | 10.69 | 8.81 | 8.56 | 0.7 | 1.3 | 1.2 | 2.0 |
| Gowalla | 0.2M | 1.0M | 13.09 | 10.26 | 9.81 | 0.9 | 1.5 | 1.4 | 2.2 |
| Skitter | 1.6M | 11.1M | 14.44 | 11.92 | 11.58 | 0.8 | 1.2 | 1.3 | 2.1 |
| DBLP | 0.2M | 1.0M | 16.21 | 11.98 | 11.23 | 0.9 | 1.4 | 1.5 | 2.3 |

Table 13: Compression rates and speeds of incomplete autoregressive shuffle coding with 16 chunks and 3 color refinement convolutions based on the AP model, on random graphs sampled from $G(n, m)$ Erdős-Rényi models with large numbers of edges $m$ up to one billion, and $n = 3/10 \cdot m$ vertices. We show the uncompressed size, meaning the ordered rate for the model it was sampled from, both in megabytes total and bits per edge. Rates and net rates are reported in bits per edge, and multi-threaded compression speeds in MB/s.

| | | Ord. Rate | | Rate | | Speed (8 Threads) | |
| Vertices | Edges | ER/MB | ER | AP | Net AP | Enc. | Dec. |
|---|---|---|---|---|---|---|---|
| 30k | 100k | 0.2 | 13.6 | 10.7 | 10.1 | 1.2 | 1.4 |
| 300k | 1M | 2.1 | 16.9 | 13.3 | 12.5 | 1.4 | 1.7 |
| 3M | 10M | 25 | 20.2 | 15.7 | 14.9 | 1.2 | 1.3 |
| 30M | 100M | 294 | 23.5 | 18.2 | 17.2 | 0.8 | 0.9 |
| 300M | 1B | 3358 | 26.9 | 20.6 | 19.5 | 0.4 | 0.6 |

