# OpenReview forum: "Practical Shuffle Coding"
_NeurIPS.cc/2024/Conference — NeurIPS 2024 poster_

### Official Review · Reviewer_tkqf · 2024-06-22

**Soundness:** 4
**Presentation:** 4
**Contribution:** 3
**Rating:** 7
**Confidence:** 3

**Summary:**

The design of compression mechanisms for compression of unordered objects is considered. The scenario models various applications of interest such as compression of unlabeled graphs and multisets. A novel recursive solution, called recursive shuffle, is introduced. An advantage of this compression mechanism, compared to previous plain shuffle method, is that it can be applied in one-shot compression scenarios. However, it is noted that since the mechanism requires computation of orbits of prefixes, it cannot be completed in polynomial time. To mitigate this, an alternative `sub-optimal' mechanism is introduced which does not jointly compress all of the order information, and handles part of the information separately. The mechanism is called incomplete recursive shuffle. Various simulations on real-world datasets are provided to compare the performance of the proposed mechanism with the state of the art methods.

**Strengths:**

- The compression problem considered in this work is of interest and is applicable to a wide range of graph compression problems.
- The method, using a recursive process that interleaves encoding and decoding steps to allow for one-shot compression, and which avoids jointly compressing all order information to reduce run-time is novel and interesting.
- The paper is well-written, and the proof arguments and explanations are complete and clear.
- The manuscript includes comprehensive numerical simulations and comparisons with the state of the art methods both in terms of compression rate and speed of compression.

**Weaknesses:**

- As mentioned in the manuscript, the recursive shuffle mechanism requires computation of the orbits of the prefixes, which cannot be solved in polynomial time. The alternative of separately compressing part of the order information is suboptimal in terms of compression rates.
-  In several of the scenarios for graph compression, the experimental results do not seem to show significant gains compared to SZIP and Plane shuffle methods. For instance, in Table 5, for one-shot compression, SZIP outperforms the proposed methods in two of the datasets and performs comparably in the other four. Furthermore, while the compression speed is significantly faster than plain shuffle, it is still below that of SZIP in several cases as shown in Table 6.

**Questions:**

- It appears from Tables 5 and 6 that the performance of SZIP is comparable and even better than the proposed methods on average. Please comment on why and in what respect if any do the proposed methods outperform SZIP?
- In Table 4, sometimes WL_1 and sometimes WL_2 hashing yields better results. I wonder if there is an intuitive explanation and a way to determine what type of hash should be used beforehand. Also, the proposed methods sometimes outperform SZIP and sometime do not. Please comment if there are specific conditions (such as graph statistics) that would cause one mechanism to outperform in different scenarios.

**Limitations:**

- The comparison with SZIP performance should be made more explicit and comprehensive in the discussions in Section 6.

---

> ### Author Rebuttal · Authors · 2024-08-07
>
> We thank reviewer tkqf for taking the time to review our paper.
>
> It will be helpful to respond to part of your second question first:
>
> > In Table 4, sometimes WL$_1$ and sometimes WL$_2$ hashing yields better results. I wonder if there is an intuitive explanation and a way to determine what type of hash should be used beforehand. Also, the proposed methods sometimes outperform SZIP and sometime do not.
>
> We thank the reviewer for pointing this out. Using more Weisfeiler Leman hashing iterations with incomplete recursive shuffle coding will always result in equal or better rates in expectation over all possible initial messages, assuming that hash collisions are rare enough, a condition practically achieved with standard 64-bit hashing methods. Our paper was misleading in this sense, and will be amended, in two ways:
>
> - As detailed in our response to reviewer Htst and briefly mentioned in Appendix C, the AP model uses a variational approximation. In the paper we failed to clarify that this can lead to the rate depending on the initial message, making it appear stochastic. We will update the paper accordingly.
> - In Table 4 we only reported each compression rate result for the AP model based on a single run. This was misleading in the sense that WL$_2$ appeared worse than WL$_1$ for some graphs. As stated in the overall response, we have now repeated each of these experiments three times with different initial message seeds, and the new results, shown in Table S1 of the supplemental page, are compatible with the prediction that WL$_2$ has a rate at least as good as WL$_1$. We now also report empirical standard deviations to characterize the stochasticity of the AP model.
>
> We will now respond to the remaining questions together:
> > Also, the proposed methods sometimes outperform SZIP and sometime do not. Please comment if there are specific conditions (such as graph statistics) that would cause one mechanism to outperform in different scenarios.
>
> > It appears from Tables 5 and 6 that the performance of SZIP is comparable and even better than the proposed methods on average. Please comment on why and in what respect if any do the proposed methods outperform SZIP?
>
> As described in the overall response, our new results in Table S1 show that incomplete shuffle coding now comfortably outperforms SZIP both with AP/WL$_1$ and AP/WL$_2$ in terms of one-shot compression rate.
>
> Our proposed methods are entropy coding methods, with optimal rates for a model that can be easily inspected, improved, and swapped out.
> Shuffle coding methods can therefore be easily adapted to specific domain models, and automatically benefit from advances in generative graph modeling. The model update mentioned above leading to the drastically improved rates from Table S1 demonstrates this capability.
>
> This is not possible for SZIP, since it is not based on an explicit probabilistic model. If its implicit model fails, there is no known method to improve it. Similarly, it is difficult to reason about the strengths of SZIP.
>
> One thing that we do know is that for large enough graphs, SZIP cannot perform significantly worse than shuffle coding with an Erdős–Rényi graph model, due to the rate upper bound discussed in Section 5,
>
> $\log\frac{1}{P_\mathrm{ER}(g)} - n\log n + O(n)$.
>
> It is hard to know how large graphs have to be for this bound to have practical significance, since the constant factor for $O(n)$ is unknown.
>
> In contrast, we can easily inspect the explicit model used by shuffle coding. We know, for example, that preferential attachment models, like the Pólya urn models (PU / AP) used in our experiments, are suitable whenever there is a ‘rich get richer’ dynamic in the generating process, meaning the rate at which new neighbors are attached is approximately proportional to the number of neighbors already present, leading to skewed neighbor count distributions (see Severo et al. 2023 for more details). This dynamic is present in many natural contexts such as social networks or web graphs, providing a possible explanation for good performance of Pólya urn models on such graphs.
>
> We will extend the discussion on SZIP in Section 6 accordingly to be more comprehensive.
>
> ### References
>
> Severo, Daniel, et al. (2023): Random Edge Coding: One-Shot Bits-Back Coding of Large Labeled Graphs." arXiv preprint arXiv:2305.09705.

---

> > ### Comment · Reviewer_tkqf · 2024-08-08
> >
> > I appreciate the authors' comprehensive response. My concerns regarding the comparison with SZIP have been fully addressed. Corrections have been made to the experimental setup which now yield consistent outcomes for WL1 and WL2 hashing. I have updated my score to reflect the improvements.

---

### Official Review · Reviewer_Htst · 2024-07-11

**Soundness:** 3
**Presentation:** 2
**Contribution:** 2
**Rating:** 4
**Confidence:** 4

**Summary:**

This paper proposes recursive shuffle coding, a general method for optimal compression of unordered objects using bits-back coding. And the paper further presents present incomplete shuffle coding, allowing near-optimal compression of large unordered objects with intractable automorphism groups. When combined, these methods achieve state-of-the-art one-shot compression rates on various large network graphs at competitive speeds.

**Strengths:**

1. Incomplete recursive methods improve the speed of processing large-scale unordered multiple sets.
2. The paper is a solid contribution with solid theoretical foundations. Introducing group theory to consider this problem is very interesting.

**Weaknesses:**

1. The two methods seem to address different issues and do not clarify their connection.
2. The symbols in the paper are too numerous, making it difficult to intuitively understand your intention.

**Questions:**

1. Could you specifically explain more details of the Autoregression PU model in Appendix C?
2. Could you clarify the connection between recursive shuffle coding and incomplete shuffle coding?

**Limitations:**

1. Less evaluation on unordered graphs with vertex and edge attributes.
2 Answer my questions Q1 and Q2.

---

> ### Author Rebuttal · Authors · 2024-08-07
>
> We thank reviewer Htst for their review, comments, and questions.
>
> ## Weaknesses
> > The two methods seem to address different issues and do not clarify their connection.
>
> Your observation is correct, the two methods address different issues. They appear together in this paper since it is convenient to use recursive shuffle coding to implement incomplete shuffle coding, and their advantages can be combined this way. Specifically, we implement incomplete shuffle coding by modifying the orbits function to return orbits of an incompletely ordered object, approximating the orbits (and automorphism group) of the underlying ordered object.
> It would be feasible for incomplete shuffle coding to be based on plain shuffle coding instead, requiring a function that returns an approximate canonization and automorphism group for a given object. While this would be an interesting direction of research, readily available graph isomorphism libraries do not provide such functionality. Therefore, we leave implementing this idea for future work and focus on the approach based on recursive shuffle coding.
>
> > Less evaluation on unordered graphs with vertex and edge attributes.
>
> As stated in the overall response, we ran additional experiments on a vast array of graph datasets featuring vertex and edge attributes, confirming that the favorable properties of our method extend to attributed graphs.
>
> ## Questions
>
> > The symbols in the paper are too numerous, making it difficult to intuitively understand your intention.
>
> While we agree that this is a technical paper, we spent great effort on visualizations to help motivate and clarify the technical concepts required for formalizing our method. In particular, we strongly encourage the reviewer to revisit the example in Tables 1 and 2, as well as Figure 1.
>
>
> > Could you specifically explain more details of the Autoregression PU model in Appendix C?
>
> Yes. We will extend Appendix C and also give a detailed explanation here.
>
> The joint model we want to approximate is the Pólya urn model from Kunze et al. (2024). Its iterative generative process starts from an empty graph, into which a fixed number of edges are inserted iteratively. The two vertices forming the next inserted edge are sampled with probabilities proportional to the current number of vertex neighbors (+1). This process favors vertices that already have many neighbors in a ‘rich get richer’ dynamic, often called ‘preferential attachment’, with a skewed distribution over vertex neighbor counts.
> Herby, self-loops and redraws are disallowed. This breaks edge-exchangeability, leading to a 'stochastic' codec, meaning that the code length depends on the initial message. Shuffle coding is compatible with such models. In this more general setting, the ordered log-likelihood term in the optimal rate of Eq. 3 is replaced with a variational `evidence lower bound' (ELBO). The discount term is unaffected. The derivations in the main text are based on the special case of exchangeable models, where log-likelihoods are exact, for simplicity. They can be generalized with little effort and new insight.
>
> We will now describe the approximate Pólya urn (AP) model that is autoregressive over graph slices. A graph slice $f_i$ can be represented as the set of edges connecting the vertex $i$ with any following vertex, i. e. a subset of $\lbrace (i, i+1), (i, i+2), … (i, n-1) \rbrace$. Any autoregressive graph model will provide a distribution over slices $f_i$, given the graph prefix $f_{[i]}$ (that comprises all previous slices). We can break down the task into two parts: Given the prefix $f_{[i]}$, predict the number of edges $k_i \in [n - i]$ within the next slice $f_i$, and then from that, predict the edge positions within the slice.
> For the Pólya urn model, the distribution for the first part, $P(k_i | f_{[i]})$, is not tractable and we instead approximate it by a truncated log-uniform distribution $Q(k_i | f_{[i]})$ where $\lfloor \log k_i | f_{[i]} \rfloor$ is (approximately) uniformly distributed within the possible range.
>
> The distribution of the second part, $P(f_i | f_{[i]}, k_i)$, is tractable with the following generative process: Given the graph of the prefix $f_{[i]}$ and the number of edges $k_i$ in the next slice $f_i$, iteratively insert $k_i$ edges, with probabilities proportional to the neighbor count + 1 of the adjacent vertices $\{ i+1, i+2, … n-1\}$ without allowing repetition (i. e. zeroing out the probabilities where edges were already inserted).
>
> We then have the slice distribution $Q(f_i | f_{[i]}) = \sum_{k_i=0}^{n-i} P(f_i | f_{[i]}, k_i) Q(k_i | f_{[i]})$, and finally the complete AP model $Q(f) = \prod_{i=0}^{n-1} Q(f_i | f_{[i]})$.
>
> The approximation of $k_i$ increases the stochasticity of the compression rates, much like disallowing self-loops/redraws. As described in the overall response, we conducted further experiments to characterize this stochasticity. We report empirical means and standard deviations of compression rates for the experiments from Table 4 with the AP model, on the supplemental page in Table S1. It shows that the resulting stochasticity is significant, highlighting that our choice of $Q(f_i | f_{[i]})$ is a crude approximation. Improving it, for example, by using information from the prefix $f_{[i]}$, is left for future work.
>
> > Could you clarify the connection between recursive shuffle coding and incomplete shuffle coding?
>
> This is answered in the first section of this response.

---

### Official Review · Reviewer_NPhs · 2024-07-12

**Soundness:** 3
**Presentation:** 4
**Contribution:** 3
**Rating:** 6
**Confidence:** 2

**Summary:**

This paper proposed an entropy coding method of  large unordered data structures. The newly proposed method allows one-shot  compression and achieves competitive speed. The experimental results demonstrate the advantages.

**Strengths:**

This paper proposed a new entropy coding method for large unordered data sets with near-optimal compression. Specially, the new method allows one-shot compression. In addition, the new method has competitive speed, which is verified by the experiments.

**Weaknesses:**

As a contribution, the proposed method in this paper can work for one-shot compression while the SOTA works can not. Hence, the applications of one-shot compression should be presented and stressed in details.

**Questions:**

What are the applications of one-shot compression?

**Limitations:**

No. The comparison with other works for common cases (other than one-shot compression) should be discussed.

---

> ### Author Rebuttal · Authors · 2024-08-07
>
> We thank reviewer NPhs for taking the time to review and comment on our paper.
>
> ### Questions
> > What are the applications of one-shot compression?
>
> Example applications include storing/transmitting large social, web, network, or compute graphs, JSON files (nested multisets), machine learning datasets, and relational database tables with many rows (multisets). Large objects of these kinds are often sent/stored separately, making one-shot compression an interesting problem. For this reason, the initial bits problem of plain shuffle coding poses a fairly substantial limitation in practice.
>
> ### Limitations
> > The comparison with other works for common cases (other than one-shot compression) should be discussed.
>
> As described and discussed in the overall response, we performed additional experiments on all TU datasets comparing our method to plain shuffle coding outside of the one-shot regime, for graphs with vertex and edge attributes. Specifically, we compress the many graphs per dataset together in sequence. The experiments clearly show that outside of the one-shot regime, incomplete recursive shuffle coding can offer dramatically faster speeds for relatively small increases in compression rate.
>
> We hope that this addresses your concerns.

---

> > ### Comment · Reviewer_NPhs · 2024-08-12
> > **Thank the authors for the response to my comments.**
> >
> > I think the rebuttal addressed my major concerns and I updated the score accordingly.

---

### Official Review · Reviewer_AUZQ · 2024-07-12

**Soundness:** 3
**Presentation:** 3
**Contribution:** 2
**Rating:** 7
**Confidence:** 3

**Summary:**

Coding of unordered structures is considered. This paper addresses the two main limitations of the entropy coding method proposed by Kunze et al. (2024): high cost of automorphism group calculation and poor compression of single unordered objects. To solve the first problem, it is suggested to approximate the object symmetries. The second problem is solved via a recursive coding, which requires an autoregressive probability model. Experiments are performed to demonstrate the merits of the proposed improvements in comparison to the original (``plain'') entropy coding method.

**Strengths:**

- The paper is well written, with all the required preliminary information included. The definitions, theorems, and proofs are mathematically rigorous. However, a great effort has also been made to facilitate the understanding of the article via various examples and clarifications, which is also commendable.
- The proposed solutions to the issues raised are elegant and might admit a generalization to arbitrary groups of automorphisms.

**Weaknesses:**

- Recursive variant of the method requires an autoregressive probability model to encode the slices. Such probabilities can be untractable or hard to compute in case of complex models. This might limit the compression capabilities of the method in the case of complex structures. For example, the considered ER and PU random graph models might yield significantly suboptimal results for complex networks, but more advanced models might also be inapplicable due to intractable conditional probabilities.
- The tradeoff between the approximation of the automorphisms group and encoding/decoding speed and compression rate is under-researched. Only two methods for approximate graph automorphisms calculation are considered.
- It seems that the method can be abstracted from the ordered/unordered objects and permutations, and can be described in terms of equivalence relations solely. This approach should unify the "plain" and "approximate" methods and slightly simplify the article. I believe that the current state of the work already requires only the slightest changes to finalize this transition.
- In the speed comparison table, the encoding speed values for SZIP are not measured on the same hardware as the rest of the experiments, but adopted from the original article.

**Questions:**

- Have you considered generalizing your method to any objects with symmetries inducing the groups of automorphisms?
- Is it possible to extend your method to non-discrete symmetries? For example, one can consider translation and rotation invariance of images and 3D objects. Such symmetries can "disappear" after the floating point discretization due to the machine precision, but it might be possible to define "approximate" automorphisms.
- Have you considered other random graph models, e.g., admitting the clustered structure?
- Is it possible to conduct more experiments with variable "accuracy" of automorphisms calculation (from very poor to almost exact)? It seems interesting to vary the rate increase given in eq. 7 and measure the change in encoding/decoding speed.
 - Repetition in Table 4: ``All measurements are in bits per edge. All measurements are in bits per edge.''
 - Is it possible to add a table, similar to Table 2, but for graph prefixes and other related concepts?

**Limitations:**

The authors briefly mention some of the limitations in the conclusion. However, I believe that the limitations related to the requirement of an autoregressive probability model should also be mentioned and discussed thoroughly.

---

> ### Author Rebuttal · Authors · 2024-08-07
>
> We thank reviewer AUZQ for their thoughtful, thorough review.
>
> ## Weaknesses
> > It seems that the method can be abstracted from the ordered/unordered objects and permutations, and can be described in terms of equivalence relations solely. This approach should unify the "plain" and "approximate" methods and slightly simplify the article. I believe that the current state of the work already requires only the slightest changes to finalize this transition.
>
> The described generalization is possible and would allow compressing elements of arbitrary quotient sets. This class of methods can be summarized as bits-back coding (Townsend et al., 2019) restricted to models with deterministic conditional distributions $P(f_\sim | g)$, which includes all shuffle coding variants.
>
> However, the existence of efficient algorithms for (approximate) orbit functions makes permutable classes something of a sweet spot in the trade-off between generality and practicality, hence the choice of framing for this paper.
>
> > In the speed comparison table, the encoding speed values for SZIP are not measured on the same hardware as the rest of the experiments, but adopted from the original article.
>
> We agree with the reviewer that having results on the same hardware would be valuable. Unfortunately, we have not found a public implementation for SZIP method, making such comparison difficult.
>
> ## Questions
> > Have you considered generalizing your method to any objects with symmetries inducing the groups of automorphisms?
>
> Yes. This generalization is straightforward, with the resulting rate being
>
> $\log \frac{1}{P(\bar{f})} = \log \frac{1}{P(f)} - \log \frac{|G|}{|\text{Aut}(f)|}$,
>
> where $G$ is the considered group, i. e. what’s $S_n$ in the paper. Hereby, the ratio inside the discount term computes the size of each coset of $\text{Aut}(f)$.
>
> All compelling applications that we found have objects that can be arbitrarily permuted. To keep this already quite technical paper more accessible, we chose to present the paper based on the most important case of a full symmetric group $G=S_n$.
>
> > Is it possible to extend your method to non-discrete symmetries?
>
> Continuous objects have infinite information content, i. e. no probability mass function can be defined over them. Therefore, they require discretization before any lossless compression method can be applied. Our method can be applied to such discretized objects, as usual. The discrete case is therefore sufficient for our practical purposes.
>
> In some cases, however, we can still generalize our rate for unordered objects to a ‘rate density’. For example, given a probability density function $p(f)$ over continuous objects f, and under the condition that the size of the cosets $c(f)$ of $\text{Aut}(f)$ is finite, we obtain a probability density function $p(\bar{f})$ over ‘unordered’ objects, via $p(\bar{f}) = p(f) \cdot c(f)$, and the corresponding ‘rate density’ $\log \frac{1}{p(\bar{f})} = \log \frac{1}{p(f)} - \log c(f)$.
>
> > Have you considered other random graph models, e.g., admitting the clustered structure?
>
> Yes. Generative modeling of graphs is an active area of research, see Zhu et al. (2022) and Maneth et al. (2015) for surveys. It is somewhat orthogonal to our work since any advance in graph modeling will allow better rates with shuffle coding. The models used in our paper have few parameters, with which we outperform competing methods. Finding and applying models that exploit more structure is a promising research direction to improve rates quickly. As explained in more detail in response to reviewer Htst, our simple Pólya models already exploit a ‘rich get richer’ generative process present in many natural contexts (such as social networks), where vertices with many neighbors have a proportionally higher probability to accumulate even more edges.
>
> > Is it possible to conduct more experiments with variable "accuracy" of automorphisms calculation (from very poor to almost exact)? It seems interesting to vary the rate increase given in eq. 7 and measure the change in encoding/decoding speed.
>
> Yes. As stated in the overall response, we ran an additional experiment with more variants of incomplete shuffle coding, reporting compression rates and speeds. For this, we chose three datasets with relatively small graphs because more WL iterations quickly become too slow on large graphs as local graph convolution features typically degenerate into global ones exponentially.
>
> The experiment shows that increasing WL iterations quickly approaches the optimal rate. In all 3 experiments, the exact optimal rate is achieved within 2 to 5 iterations. This corresponds to the fact that the Weisfeiler-Leman graph isomorphism test is effective on practical graphs.
>
> > Repetition in Table 4: "All measurements are in bits per edge. All measurements are in bits per edge."
>
> Thanks, this will be fixed!
>
> > Is it possible to add a table, similar to Table 2, but for graph prefixes and other related concepts?
>
> This is a fantastic idea, and we will include it in the final version of the paper.
>
> ## Limitations
> > I believe that the limitations related to the requirement of an autoregressive probability model should also be mentioned and discussed thoroughly.
>
> Agreed, will do! We want to emphasize here that we propose joint shuffle coding to mitigate exactly this restriction. Additionally, recent work on autoregressive models is promising (see Kong et al., 2023), and our results show that such models can lead to competitive rates even with few parameters.
>
> ## References
> Zhu, Yanqiao et al. (2022): A survey on deep graph generation: Methods and applications. Learning on Graphs Conference. PMLR.
>
> Maneth, Sebastian et al. (2015): A survey on methods and systems for graph compression. arXiv preprint.
>
> Townsend, James et al. (2019): Practical lossless compression with latent variables using bits back coding. ICLR.
>
> Kong, Lingkai, et al. (2023): Autoregressive diffusion model for graph generation. ICML. PMLR.

---

> > ### Comment · Reviewer_AUZQ · 2024-08-11
> >
> > Thank you for answering my questions and addressing the raised concerns.
> >
> > I am mostly satisfied with the answers and overall effort of the authors to improve their work.
> > I would like to additionally commend the new experimental results the authors provide.
> > However, I think that the results presented in the Figure S1 should be plotted in the log or log-log scale
> > to better distinguish the converging lines.
> > Additionally, in my opinion, the limitation regarding the complexity of obtaining conditional probabilities
> > (required for encoding the slices in the recursive algorithm) for complex network models is still somewhat underdiscussed.
> >
> > In conclusion, I think that the paper is well-written, and the experimental evaluation is strong.
> > Recognizing the results presented during the rebuttal, I would like to increase my score to "7: Accept".
> > I still believe that the paper lacks some generalization,
> > and some limitations can be discussed more explicitly,
> > but these issues can be considered minor.

---

### Author Rebuttal · Authors · 2024-08-07

We are delighted that all reviewers agreed on the good soundness of the paper, with multiple pointing out its mathematical rigor, and most reviewers appreciate its presentation.

Based on reviewers’ requests, we are excited to report striking new experimental results on our supplemental page, described here, and discussed further in the individual responses:

- Table S1 updates Table 4 from the paper. We found that for incomplete recursive shuffle coding with the autoregressive Pólya urn (AP) graph model, to compress edge counts $k_i$ in slices, using a simple approximation based on a discretized Pareto distribution (specifically, $Q(k_i) \propto \frac{1}{2} k_i^{-2}$ for $k_i>1$ with $Q(k_i=0)=Q(k_i=1)=\frac{1}{3}$) leads to much-improved rates over the previously used log-uniform distribution. As a result, our method now comfortably outperforms SZIP in terms of one-shot compression rate for both the AP/WL$_1$ and AP/WL$_2$ configurations. This also improves all results in Table 5, which we will update in the final version.
- We now repeat these AP experiments with varying seeds, and report means and empirical standard deviations. As explained further in the responses below, this highlights the stochastic nature specific to the AP model and supports the prediction that more WL iterations in practice lead to better or equal expected rates. Previous results for AP in Table 4 were misleading in these two respects.
- In Table S2, we compare incomplete recursive with plain shuffle coding on all the 136 TU graph datasets (Morris et al., 2020) in six categories, ranging from molecules to social networks, with the majority of these datasets featuring vertex and edge attributes (see [chrsmrrs.github.io/datasets/docs/datasets](https://chrsmrrs.github.io/datasets/docs/datasets/) for details). Our method leads to a dramatic speedup over plain shuffle coding on some of these datasets, with a minimal increase in compression rate across all datasets, showing that these favorable properties extend to graphs with attributes. Maximum-likelihood parameters for a categorical distribution over vertex and edge attributes are inferred for and coded along with each dataset, as done and described in Kunze et al. 2024.
- Figure S1 shows how the compression rate and speed varies with the automorphism group's approximation level, on three such graph datasets. This experiment further confirms that in practice, a low number of WL iterations is sufficient to get very close to the optimal rate.

**Crucially, the new results show that our method now achieves state-of-the-art rates in one-shot graph compression, as well as competitive rates and speeds on graphs with attributes, and explore the method’s rate-speed trade-off. We believe these results significantly strengthen the paper, particularly in the 'contribution' category.** We hope that the reviewers agree, find our responses satisfactory, and that their concerns are being fully addressed.

### References

Morris, Christopher et al. (2020): TUdataset: A collection of benchmark datasets for learning with graphs. Graph Learning and Beyond workshop, ICML. [chrsmrrs.github.io/datasets](https://chrsmrrs.github.io/datasets/)

Kunze, Julius et al. (2024): Entropy Coding of Unordered Data Structures. ICLR.

---

> ### Author Response · Authors · 2024-08-13
>
> We thank all reviewers for their responses to our rebuttal. We are thrilled that all responses so far are very positive and indicate satisfaction with our rebuttal, with ratings raised by 1 to 2 grades each. We fully agree with reviewer AUZQ's additional feedback, which we will implement in the paper.
>
> We also hope that our rebuttal has fully addressed reviewer Htst's questions and concerns, and would much appreciate their response before the discussion period ends within the next 24 hours.

---

### Decision · Program_Chairs · 2024-09-25

**Decision:**

Accept (poster)

**Comment:**

The paper proposes recursive shuffle coding, a general method for comparison of unordered data structures (multisets, unlabeled graphs, etc.) based on bits-back coding. The paper improves on recent previous works that utilize similar techniques for compressing unordered structures. Recursive shuffle coding, unlike the previous "plain" shuffle coding, can be applied in one-shot compression scenarios (where a single unordered object needs to be compressed). Moreover, the paper proposes incomplete shuffle coding, which mitigates the fact that shuffle coding requires computing the automorphism group (which can be computationally intractable). The methods achieve state-of-the-art compression rates on several datasets.

The reviewers mostly agree that this paper provides meaningful improvements over previous shuffle coding approaches. As compression of unordered data structures has a wide range of applications, this is a meaningful contribution.